# LEAP: Zone-Aware MCTS for LLM Self-Speculative Decoding

**Leiquan Zheng** [1]   **Yuan Liu** [1]

## Abstract

Self-speculative decoding accelerates LLM inference by using a lightweight draft model for generation and a target model for verification, where the draft model is constructed by a subset of the target model's layers, and the key challenge lies in layer configuration strategies. To address this challenge, we propose LEAP, a plug-and-play approach that formulates and optimizes the draft model construction problem as a sequential decision-making process by Monte Carlo Tree Search (MCTS). To navigate the prohibitive search space of deep LLMs, we leverage two empirical observations: (i) the prefilling-derived redundancy information remains informative during decoding, and (ii) the layer redundancy exhibits zone-wise characteristics. These observations enable a structured search space through zone partitioning and layer grouping, which serves as an inductive bias to facilitate efficiency of MCTS. Experimental results show that LEAP achieves a speedup of $1.7\times \sim 2.0\times$ for LLM inference. We release our code in https://github.com/leiquanzheng/LEAP.

## 1. Introduction

Large language models (LLMs), such as GPT, DeepSeek and LLaMA (Achiam et al., 2023; Touvron et al., 2023; Liu et al., 2024), have achieved strong performance across tasks (Kaplan et al., 2020; Brown et al., 2020). However, their deployment remains constrained by high latency and computation overhead from autoregressive generation and massive model parameters.

Speculative decoding (Leviathan et al., 2023; Xia et al., 2023) accelerates LLM inference by incorporating a small draft model and a large target model. At each iteration, it generates multiple draft tokens by the draft model sequentially and then verifies them by the target model in parallel. However, speculative decoding critically depends on the alignment between the draft and target models. So many variants (Li et al., 2024a;b; 2026; Cai et al., 2024) construct draft models by introducing extra modules or training, which incurs substantial time and computation overhead, and limits their generalization across models and scenarios.

Self-speculative decoding (Zhang et al., 2024; Elhoushi et al., 2024) provides a promising paradigm where the draft model is constructed by a subset of the target model's layers, enabling training-free acceleration as well as high alignment because of the shared model parameters and KV cache. The optimization of the layer configuration strategy is the key issue in self-speculative decoding. To this end, the existing methods (Zhang et al., 2024; Xia et al., 2025; Chen et al., 2025) optimize the draft model by skipping a fixed number of target model's layers during inference, which lacks adaptability across different tasks. Moreover, faced with a large search space and computational overhead, they optimize through accumulated information across multiple instances or reducing the frequency of search, which leads to delayed optimization benefits.

To address these limitations, we propose LEAP, a plug-and-play self-speculative decoding framework that formulates draft model construction as a sequential decision-making process and optimizes by Monte Carlo Tree Search (MCTS). LEAP leverages MCTS to directly optimize inference speedup in speculative decoding by incrementally exploring layer configurations across a limited number of iterations. At each iteration, LEAP evaluates different layer configurations to assess the impact of per-layer-group actions on speedup, rather than per-layer actions, and updates the optimization direction accordingly. This design enables fine-grained and efficient search, allowing potential configurations to be rapidly identified and subsequently explored. Importantly, speculative decoding can provide immediate and informative speedup feedback for each evaluated configuration in MCTS, acting as the simulation within the MCTS process and enabling on-the-fly optimization with negligible overhead.

However, directly applying MCTS to the deep LLMs also presents the challenge of prohibitively large search space.

[1]School of Electronic and Information Engineering, South China University of Technology, Guangzhou 510641, China. Correspondence to: Yuan Liu <eeyliu@scut.edu.cn>.

*Proceedings of the 43rd International Conference on Machine Learning*, Seoul, South Korea. PMLR 306, 2026. Copyright 2026 by the author(s).

To make MCTS more tractable, we leverage two key empirical observations: (i) layer redundancy information derived during the prefilling stage remains informative throughout subsequent decoding steps, as the redundancy metrics computed at prefilling and decoding stages are strongly and stably correlated, allowing the redundancy to be estimated once during prefilling and reused during decoding; (ii) layer redundancy exhibits zone-wise characteristics, where early zone primarily contributes to feature transformation, final zone plays an important role in acceptance improvement, while middle zone exhibits relatively limited contribution to both. Based on these insights, LEAP utilizes prefilling-derived redundancy information to initially partition layers into zones that share zone-specific MCTS actions, and further groups layers within each zone. Notably, zones and groups are determined during the prefilling stage and kept fixed, leveraging the stability of redundancy information across different steps. This structured search space serves as an inductive bias for MCTS, enabling efficient exploration under a limited search budget.

Our main contributions are summarized as:

- We introduce LEAP, a plug-and-play self-speculative decoding framework that optimizes the draft model construction by MCTS. By incrementally exploring complete layer configurations, LEAP enables fine-grained and efficient optimization under a limited search budget.

- We reveal the strong correlation of redundancy information across inference stages, as well as its zone-wise characteristics. These observations motivate redundancy-aware zone partitioning of LLM, and provides a structured and reduced search space for MCTS.

- Extensive experiments across scenarios demonstrate that LEAP consistently achieves a speedup of $1.7\times \sim 2.0\times$ without any additional module or training, while preserving the original distribution of the target model.

**Conflict of Interest Disclosure.** The authors declare that they have no financial interests or personal relationships that could have influenced the work reported in this paper.

## 2. Related Work

**Training-based speculative decoding.** Many methods improve draft-target alignment by introducing trainable draft modules. For example, EAGLE (Li et al., 2024a;b; 2026) trains a transformer decoder layer as the drafter, while follow-up works enhance alignment via on-policy training or cross-attention (Zhang et al., 2025; Zimmer et al., 2025). However, these methods incur non-trivial training and computation overhead, limiting generalization across models and deployment scenarios.

**Self-speculative decoding.** Existing self-speculative decoding methods differ significantly in optimization granularity. Self-SD (Zhang et al., 2024) and SWIFT (Xia et al., 2025) formulate layer selection as a configuration-level problem and apply offline or on-the-fly Bayesian optimization, respectively. They treat skipped-layer selection as a black-box problem optimized by matchness or speedup, which prevents explicit reasoning about layer-level contributions and leads to coarse-grained exploration. In contrast, CLaSP (Chen et al., 2025) adopts a heuristic token-level skipping strategy based on single-token feature alignment. While enabling instance-specific adaptation, such strategies may not capture redundancy information specific to speculative decoding and can be affected by residual connections. Moreover, these methods typically fix the number of skipped layers as a hyperparameter, restricting their adaptability.

Unlike these methods, LEAP explores how layer-group actions affect inference speedup, enabling dynamic updates of layer configurations during speculative decoding and incrementally constructing an effective draft model. LEAP also adaptively determines the number of skipped layers based on achieved speedup.

## 3. Empirical Observations

This section presents two empirical observations that provide supporting evidence for our design.

### 3.1. Prefilling-Derived Redundancy for Decoding

During prefilling stage, the entire input context is processed by LLM, allowing us to obtain the intermediate features and probability distributions with negligible overhead. Based on these outputs, we define two metrics $R_l$ and $\Delta \bar{L}$ to represent each layer's redundancy, which will be detailed in Section 3.2. However, as decoding proceeds, the semantic content of the sequence may change, raising the concern that the redundancy of each layer may also change, so prefilling redundancy information may not be able to keep informative through decoding steps.

To address this concern, for each instance, we compute two redundancy metrics above during prefilling step and at multiple decoding steps $\{2, 4, 8, 16, 32, 64\}$ (prefilling step is the first step). We then measure the correlation between metrics at different steps by Spearman's rank correlation coefficient $\rho$.

As shown in Figure 1, metric $R_l$ obtained from prefilling step exhibits strong and stable correlation with those at decoding steps, while the correlation of metric $\Delta \bar{L}$ remains slightly lower yet still strong near 0.7. More importantly, the correlation of these two metrics maintain stable as decoding step increases, demonstrating that prefilling-derived redundancy information can serve as a reliable approximation

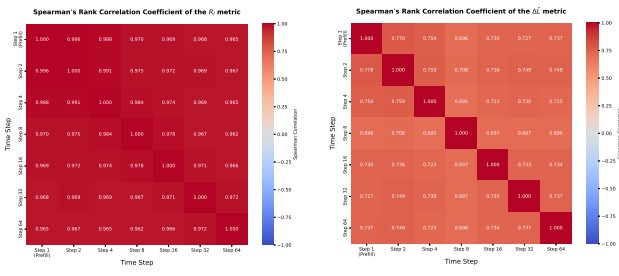

*(a)* Correlation of $R_l$ across steps.

*(b)* Correlation of $\bar{L}$ across steps.

*Figure 1.* Correlation of two redundancy metrics across steps. Each cell represents the correlation between steps, with darker colors indicating stronger correlation

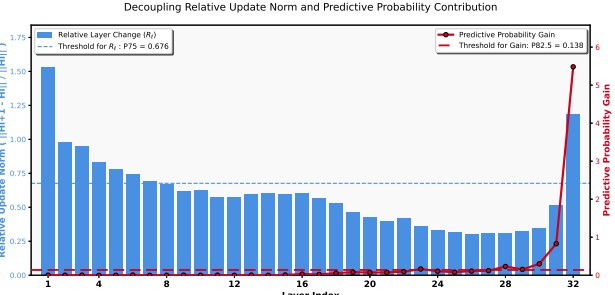

*Figure 2.* Two redundancy metrics of all layers.

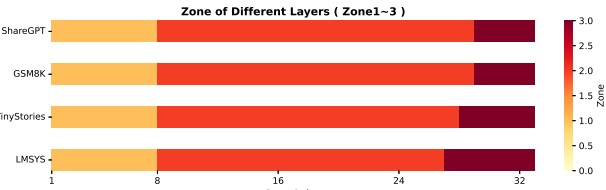

*Figure 3.* Zone-wise characteristics across tasks.

of redundancy for decoding. Consequently, it is sufficient to compute redundancy once during the prefilling stage, without repeated estimation at subsequent decoding steps.

### 3.2. Zone-Wise Redundancy Characteristics

To characterize layer redundancy more comprehensively, we consider two complementary metrics that capture feature transformation and acceptance improvement, respectively. The first metric $R_l$ measures the relative transformation of output feature $\mathbf{h}_{\text{out}}^{(l)}$ of layer $l$ relative to its input feature $\mathbf{h}_{\text{in}}^{(l)}$:

$$R_l = \frac{\left\| \mathbf{h}_{\text{out}}^{(l)} - \mathbf{h}_{\text{in}}^{(l)} \right\|_F}{\left\| \mathbf{h}_{\text{in}}^{(l)} \right\|_F}. \quad (1)$$

The second metric $\Delta\bar{L}$ captures redundancy by measuring the change in average acceptance length between adjacent layers. To simulate speculative decoding, we view each layer and all its previous layers as a single draft model and the full LLM as the target model. For the last $L$ tokens output during the prefilling stage, we greedily sample corresponding output token $x_i$ with probability $q_i$ from $L$ output distributions, then derive its corresponding probability $p_i$ from the full LLM's outputs, and calculate the acceptance rate of each token $\alpha_i = \frac{p_i}{q_i}$ and the acceptance length of each layer $\bar{L}_t$ by cumulatively multiplying the acceptance rates and summing them:

$$\bar{L}_t = \sum_{k=1}^{L} \prod_{i=1}^{k} \alpha_i. \quad (2)$$

Then we compute the metric $\Delta\bar{L}_t = \bar{L}_t - \bar{L}_{t-1}$ to quantify the change in the proximity of layer $t$ to the final prediction.

Together, these metrics provide a complementary view of layer redundancy during the prefilling stage. However, they characterize redundancy from different perspectives and for layers at different positions. Specifically, $R_l$ is more

indicative of redundancy in earlier layers, where feature transformation dominates, while $\Delta\bar{L}$ reflects acceptance improvement and provides more informative signals for layers closer to the output.

We calculate these two metrics on multiple tasks and use the $75^{\text{th}}$ and $82.5^{\text{th}}$ percentiles as the respective thresholds. Figure 2 shows that $R_l$ gradually decreases with deeper layers and suddenly increases at the last two layers, and only the shallow layers have $R_l$ values above the threshold. In contrast, $\Delta\bar{L}$ increases sharply in the final layers, indicating that they progressively converge toward the final answer. The distinct layer-wise trends of these two metrics together with their respective thresholds, jointly support a partition of layers into different zones.

Based on redundancy information, layers can be partitioned into zones as shown in Figure 3, where three different colours indicate three different zones. Early zone exhibits low redundancy on $R_l$, indicating their foundational role in processing input features. Skipping them may risk degrading the understanding of input features, making them low-return targets for inference acceleration. Middle zone shows relatively low $R_l$ and low $\Delta\bar{L}$, suggesting that skipping these layers may incur limited impact on the final output. Final zone exhibits high $\Delta\bar{L}$, indicating that these layers are closer to the final prediction and therefore should be preserved to maintain alignment with the target model.

In summary, layer redundancy exhibits zone-wise characteristics along model depth. Based on these empirical observations, we use $R_l$ to distinguish early and middle zones, and $\Delta\bar{L}$ to separate middle and final zones, enabling zone-specific layer configuration strategies in MCTS.

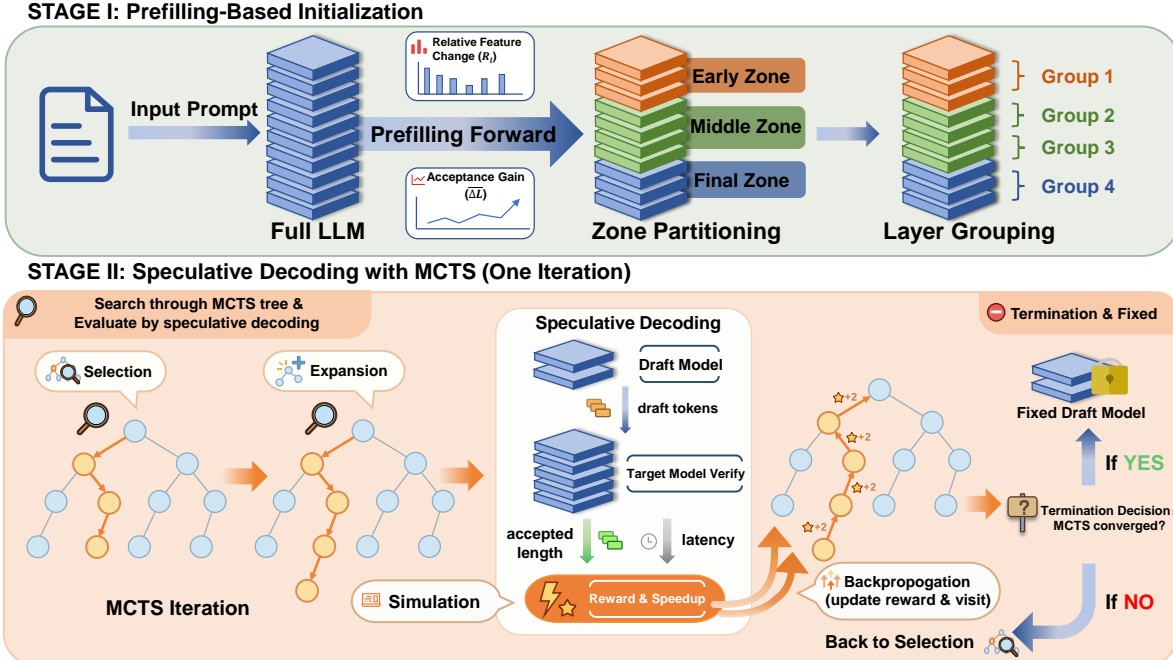

*Figure 4.* **STAGE I** illustrates the prefilling-based initialization in LEAP. After a forward pass of the input through the full LLM, LEAP derives two metrics $R_l$ and $\Delta \bar{L}$ based on intermediate features and probability distributions. Then LEAP leverages these two metrics to perform zone-wise partitioning and layer grouping. **STAGE II** illustrates one speculative decoding iteration with MCTS in LEAP. At each iteration, MCTS selects and expands a partial configuration, which is then evaluated via speculative decoding with a draft model constructed by the rollout policy. The corresponding speedup and reward is leveraged for backpropagation. Once satisfying the termination criterion, the draft model is fixed for the remaining iterations; otherwise, the above MCTS procedure continues.

## 4. Methodology

We propose a novel self-speculative decoding framework that incrementally constructs the draft model through MCTS, enabling lossless LLM acceleration without offline optimization or training.

As illustrated in Figure 4, our approach consists of two key stages: (1) prefilling-based initialization (§ 4.1), which organizes layers into redundancy-aware zones and groups based on prefilling-derived redundancy information, and (2) joint MCTS and speculative decoding (§ 4.3), which explores potential layer configurations alongside speculative decoding and fixes the optimal-speedup configuration for remaining inference acceleration. The pseudo codes of these two stages are detailed in Appendix A.

### 4.1. Initialization Based on Redundancy Information

Motivated by the empirical observations in Section 3.1 and 3.2, we compute two redundancy metrics $R_l$ and $\Delta \bar{L}$ once during the prefilling stage. Building on the complementary nature of these metrics analyzed in Section 3.2, we combine them to obtain a redundancy estimation applicable across decoding steps.

Based on the redundancy estimation, we partition layers into

early, middle, and final zones. Layers in the early zone are foundational so are constrained to execute, effectively reducing the MCTS search space by preventing aggressive skipping strategies, whereas layers in the middle and final zones allow a respective richer set of actions. Within each zone, consecutive layers are further merged into groups which serve as the basic decision units in MCTS. This grouping further reduces the search space, enabling tractable exploration under a limited search budget. The detailed grouping procedure is provided in Appendix C.

Notably, the initialization based on prefilling-derived redundancy information does not affect the final layer configuration strategy; instead, it guides MCTS toward promising strategies. All skip/execute/repeat actions for layer groups are selected online, allowing layer configuration strategy to be adaptively adjusted based on real-time feedback.

### 4.2. MCTS for Adaptive Layer Configuration

After initializing redundancy-aware zones and groups, we formulate draft model construction as a sequential decision-making process and leverage MCTS within limited search iterations to adaptively explore layer configuration strategies.

**Search space.** We define the search space in terms of layer group rather than individual layer. Given a model with $M$ layers partitioned into $m$ groups $\mathcal{G} = \{G_1, \ldots, G_m\}$, each complete configuration selects a single action for every group, drawn from the zone-specific action set. All layers within a group share the same action. This design reduces the exponential search space from $3^M$ to an upper bound of $3^m$, making the search tractable under limited iterations.

**State.** A state $S_t$ represents a partial layer configuration up to the first $t$ groups, denoted as $S_t = (a_1, a_2, \ldots, a_t)$, where $a_k$ is the action selected for group $G_k$. The initial state $S_0 = \emptyset$ represents the root with no actions selected. A state is considered *terminal* when $t = m$, indicating that a complete layer configuration is fully specified.

**Actions.** The available actions for each group are determined by its zone. Specifically, groups in the early zone only allow `execute`, groups in the middle zone allow `execute` and `skip`, and groups in the final zone allow `execute` and `repeat`. The `repeat` action is defined at the layer-group level: for a group $(A, B)$, it executes the group once more as $(A, A, B, B)$. We allow at most one repetition per group to limit the additional computation introduced by repeated execution. The middle zone focuses on skipping redundant layers, while the final zone permits repetition to compensate for aggressive skipping. These zone-aware constraints assign appropriate actions to layers based on redundancy, further reducing the effective search space from $3^m$ to fewer than $2^m$.

**Transition.** The transition updates the state by appending the selected action of the subsequent group to the partial configuration, without modifying model parameters or layer ordering.

**Simulation.** For a non-terminal state that specifies actions up to the first $n$ groups ($n < m$), the actions for the remaining groups $\{G_{n+1}, \ldots, G_m\}$ are selected using a heuristic rollout policy under zone-specific action constraints, thereby synthesizing a complete configuration. During MCTS simulation, this configuration, which constructs the draft model, is evaluated through speculative decoding. We define the reward as a function of the inference speedup:

$$\text{Speedup} = \frac{\tau \cdot T_{\text{baseline}}}{T_{\text{draft}} + T_{\text{verify}}}, \quad (3)$$

where $\tau$ is the number of generated tokens, $T_{\text{baseline}}$ is the baseline latency, and $T_{\text{draft}}$ and $T_{\text{verify}}$ are the draft and verification latency each iteration, respectively. Then we compute the reward:

$$R = f(\text{Speedup}), \quad (4)$$

where $f(\cdot)$ is a monotonic function that assigns positive rewards to high-speedup configurations and penalties to configurations that slow down inference.

**Search Objective.** We define the UCB score for state selection:

$$\text{UCB}(S) = \underbrace{\frac{Q(S)}{v(S)}}_{\text{Exploitation}} + \underbrace{c\sqrt{\frac{\ln V}{v(S)}}}_{\text{Exploration}} - \underbrace{\lambda \mathcal{L}(r(S))}_{\text{Depth Penalty}}. \quad (5)$$

Here, $Q(S)$ and $v(S)$ denote the accumulated reward and visit count of state $S$, respectively; $V$ is the total number of simulations, $c$ balances exploration, and $\lambda$ weights a soft depth penalty $\mathcal{L}(r(S))$ based on the layer retention ratio $r(S)$, which serves solely as a regularization and does not fix the final depth which is determined adaptively by MCTS.

### 4.3. Joint MCTS and Speculative Decoding

Although both our method and prior approaches operate on a full LLM, existing methods face challenges in both search capability and efficiency. In contrast, our method directly optimizes the inference speedup by incrementally exploring complete layer configurations with MCTS, enabling fine-grained and efficient search under a limited search budget.

More importantly, MCTS is particularly suited for speculative decoding. It naturally evaluates complete layer configurations through real-time inference speedup, providing informative and immediate reward for MCTS. Leveraging the reward together with the UCB score, MCTS can efficiently identify promising layer configurations within limited number of iterations. Moreover, since MCTS does not impose rigid constraints on layer configuration, it enables flexible selections of `skip`, `execute`, and `repeat` for layer groups, and can be performed alongside speculative decoding with negligible overhead.

At each iteration, a candidate layer configuration is completed from the current partial configuration via a heuristic rollout policy, which assigns actions to the remaining layer groups. Based on this configuration, the draft model $\mathcal{M}_D$ is constructed to generate draft tokens. Importantly, the cached key-value pairs of the target model $\mathcal{M}_T$ can be reused by $\mathcal{M}_D$, reducing distributional mismatch while also reducing computation and memory overhead. The generation length and execution time of each iteration are used to compute the inference speedup, which serves as the reward and is back-propagated through the MCTS tree. As decoding proceeds, MCTS incrementally refines group-wise actions based on speedup feedback, exploring potential layer configurations and progressively constructing an effective draft model for inference acceleration.

Once the maximum number of MCTS iterations is reached

or the best candidate configuration remains unchanged over multiple iterations, MCTS terminates. The optimal-speedup configuration is then fixed to construct the draft model for remaining inference, eliminating further search overhead. This termination ensures that the cost of MCTS is negligible over the whole speculative decoding process.

### 4.4. Token Tree Construction and Lossless Verification

Our method constructs a draft token tree following tree-based approaches (Miao et al., 2024; Svirschevski et al., 2024). The token tree has a fixed length $L$ and width $W$, where each layer only retains $W$ tokens with top probabilities to guarantee the quality. Tree construction enables more efficient utilization of draft computation and improves acceptance rate. These benefits are further amplified in our method by reusing target model's KV cache during draft generation.

For verification, we adopt the standard lossless verification (Leviathan et al., 2023), ensuring that the final output distribution remains identical to that of the target model.

## 5. Experiments

### 5.1. Experiment Setups

**Implementation details.** We mainly evaluate LEAP on LLaMA-3-8B and LLaMA-3-70B (Grattafiori et al., 2024) across various evaluation datasets including GSM8K (Cobbe et al., 2021), MTBench (Zheng et al., 2023), CNN/DailyMail (CNN/DM) (Nallapati et al., 2016), Natural Questions (Kwiatkowski et al., 2019) and WMT14 DE-EN. The maximum generation lengths for all datasets are 1024. Following prior work, we adopt speculative sampling (Leviathan et al., 2023) as our lossless acceptance strategy with a batch size of 1. More detailed setups about our initialization and MCTS are provided in Appendix B.3.

**Baselines.** In our main experiments, we compare LEAP to two plug-and-play self-speculative decoding methods: SWIFT (Xia et al., 2025) and CLaSP (Chen et al., 2025), both of which adaptively adjust layer configuration strategies during inference. This comparison focuses on different ways to construct the draft model within the self-speculative decoding paradigm, thereby enabling a fair evaluation of construction strategies. Other speculative decoding methods are excluded because they either adopt fundamentally different drafting paradigms, or because they require additional modules or training, which limits their generalizability. Baseline configurations are detailed in Appendix B.3.

**Evaluation metrics.** We report two widely used metrics for LEAP evaluation: mean generated length $M$ and wall-time speedup ratio compared to vanilla autoregressive decoding. Because we adopt the lossless verification which preserves the target model's output distribution, we solely focus on inference speedup instead of generation quality.

The code is accessible via

### 5.2. Main Results

Table 1 presents the comparison between LEAP and two other methods across multiple datasets, models, and sampling temperatures.

Across all scenarios, LEAP consistently achieves higher speedup and longer generation length than other approaches. Under greedy decoding, LEAP consistently achieves speedup close to or exceeding $2.0\times$, substantially outperforming prior methods whose speedup ranges from $1.1\times$ to $1.5\times$. When sampling temperature $T = 1$, LEAP presents a clear advantage, achieving approximately $1.7\times$ speedup, while other methods have a pronounced degradation.

LEAP achieves higher acceleration by enabling more effective and targeted exploration during inference. Prefilling-derived zone partitioning and layer grouping facilitates a structured search space. MCTS then incrementally explores and evaluates complete layer configurations based on real-time speedup feedback. In contrast, prior methods rely on redundancy signals that are either insufficiently informative or correlated with inference speedup, which limits their ability to adapt layer configurations and results in shorter generation length and lower overall speedup.

The speedup advantage is more pronounced on larger model LLaMA-3-70B which exhibits stronger layer redundancy. This highlights our approach's ability to consistently explore potential configurations across different model scales, leading to robust speedup improvement.

In summary, our method outperforms other approaches in different scenarios with no need of offline optimization or task-specific tuning, which demonstrates its immediate effectiveness and generalizability.

### 5.3. Ablation Studies

To investigate the speedup contribution of each component in LEAP, we conduct the ablation study. As shown in Table 2, we examine the impact of removing (1) zone partitioning (*w/o zone*), (2) layer grouping (*w/o group*), and (3) MCTS (*w/o MCTS*) independently.

**Effect of zone partitioning.** Removing zone partitioning leads to a uniform action set across all layer groups, rather than zone-specific actions, which causes the overall speedup to drop from $2.06\times$ to $1.67\times$. Without the structural prior provided by zone partitioning, MCTS wastes the search

*Table 1.* Comparison between LEAP and other plug-and-play self-speculative decoding methods. We report the mean generation length $M$ and wall-time speedup ratio on different datasets and models when Temperature=0 and Temperature=1.

| Models | Methods | GSM8K | | MT-Bench | | CNN/DM | | NQ | | WMT14 | | Overall Speedup |
|---|---|---|---|---|---|---|---|---|---|---|---|---|
| | | $M$ | Speedup | $M$ | Speedup | $M$ | Speedup | $M$ | Speedup | $M$ | Speedup | |
| | | | | | Temperature = 0 | | | | | | | |
| LLaMA-3-8B | Vanilla | 1.00 | 1.00× | 1.00 | 1.00× | 1.00 | 1.00× | 1.00 | 1.00× | 1.00 | 1.00× | 1.00× |
| | SWIFT | 2.56 | 0.86× | 5.70 | 1.48× | 5.40 | 1.34× | 2.61 | 1.00× | 2.53 | 0.96× | 1.13× |
| | CLaSP | 1.86 | 1.20× | 2.10 | 1.30× | 1.91 | 1.22× | 2.04 | 1.25× | 2.25 | 1.36× | 1.27× |
| | **LEAP** | 4.66 | **2.12×** | 4.63 | **2.07×** | 4.44 | **1.87×** | 4.71 | **2.09×** | 4.71 | **2.11×** | **2.05×** |
| LLaMA-3-70B | Vanilla | 1.00 | 1.00× | 1.00 | 1.00× | 1.00 | 1.00× | 1.00 | 1.00× | 1.00 | 1.00× | 1.00× |
| | SWIFT | 2.24 | 1.40× | 2.74 | 1.55× | 2.14 | 1.31× | 2.20 | 1.43× | 1.92 | 1.35× | 1.41× |
| | CLaSP | 3.06 | 1.54× | 3.17 | 1.56× | 3.15 | 1.39× | 3.08 | 1.49× | 3.28 | 1.62× | 1.52× |
| | **LEAP** | 4.95 | **2.00×** | 4.81 | **2.00×** | 4.93 | **1.81×** | 4.52 | **1.86×** | 4.84 | **2.03×** | **1.94×** |
| | | | | | Temperature = 1 | | | | | | | |
| LLaMA-3-8B | Vanilla | 1.00 | 1.00× | 1.00 | 1.00× | 1.00 | 1.00× | 1.00 | 1.00× | 1.00 | 1.00× | 1.00× |
| | SWIFT | 2.34 | 0.78× | 2.04 | 0.79× | 1.70 | 0.64× | 1.88 | 0.83× | 1.91 | 0.78× | 0.76× |
| | CLaSP | 1.71 | 1.06× | 1.80 | 1.02× | 1.83 | 1.09× | 1.80 | 1.03× | 1.73 | 1.09× | 1.06× |
| | **LEAP** | 4.41 | **1.86×** | 3.97 | **1.58×** | 4.00 | **1.69×** | 3.99 | **1.60×** | 3.85 | **1.66×** | **1.68×** |
| LLaMA-3-70B | Vanilla | 1.00 | 1.00× | 1.00 | 1.00× | 1.00 | 1.00× | 1.00 | 1.00× | 1.00 | 1.00× | 1.00× |
| | SWIFT | 2.48 | 1.08× | 2.52 | 1.04× | 2.62 | 1.02× | 2.84 | 1.26× | 2.58 | 1.10× | 1.10× |
| | CLaSP | 2.79 | 1.26× | 2.66 | 1.19× | 2.88 | 1.16× | 2.52 | 1.09× | 2.56 | 1.11× | 1.16× |
| | **LEAP** | 4.58 | **1.82×** | 4.41 | **1.81×** | 4.66 | **1.65×** | 3.89 | **1.59×** | 4.35 | **1.80×** | **1.73×** |

budget on ineffective strategies which skip early layers, resulting in limited acceleration. This demonstrates that zone partitioning is effective to guide MCTS toward high-reward configurations and ensure efficiency in speculative decoding.

**Effect of layer grouping.** Disabling layer grouping results in a speedup drop to 1.63×. Without grouping, the basic decision unit of MCTS changes from layer group to the individual layer, which significantly expands the search space. Within a limited search budget, this enlarged search space prevents MCTS from exploring more potential configurations. The results confirm that grouping is essential for reducing the search space to enable MCTS to explore high-speedup configurations efficiently.

**Effect of MCTS search.** We replace MCTS with a random search within the same number of search iterations. We independently sample each iteration's complete configuration based on the target layer retention ratio and finally select the speedup-optimal one. As observed, the acceleration degrades to 1.56×. Although zone partitioning and layer grouping provide a structured search space, the absence of MCTS fails to leverage real-time speedup feedback to update optimization direction effectively and explore potential configurations. This confirms that MCTS is critical for feedback-driven and adaptive exploration. By iteratively refining group-wise actions based on the speedup feedback, MCTS efficiently explores potential layer configurations and incrementally constructs the effective draft model.

*Table 2.* Ablation study on different components in LEAP.

| Methods | GSM8K | MT-Bench | WMT14 | Overall Speedup |
|---|---|---|---|---|
| **LEAP** | 2.12× | 2.07× | 2.11× | 2.10× |
| *w/o zone* | 1.78× | 1.64× | 1.61× | 1.67× |
| *w/o group* | 1.89× | 1.48× | 1.53× | 1.63× |
| *w/o MCTS* | 1.62× | 1.55× | 1.51× | 1.56× |

**Summary.** Overall, the ablation results indicate that three components contribute to speedup in an complementary manner. Zone partitioning facilitates early-stage search, layer grouping reduces search space to motivate effective exploration, and MCTS leverages real-time feedback to adaptively identify potential layer configurations. Consequently, removing any single component leads to a consistent degradation in speedup.

### 5.4. In-depth Analysis

5.4.1. ANALYSIS OF EFFICIENCY AND PERFORMANCE

Figure 5 illustrates the comparison of per-instance and average speedup between LEAP and SWIFT (Figure 5a), and between LEAP and CLaSP (Figure 5b), where instances are ordered along the x-axis, scattered points represent per-instance speedup and solid lines represent the average speedup.

**Efficiency.** Our method achieves high speedup from the beginning and quickly stabilizes at a high level. In contrast,

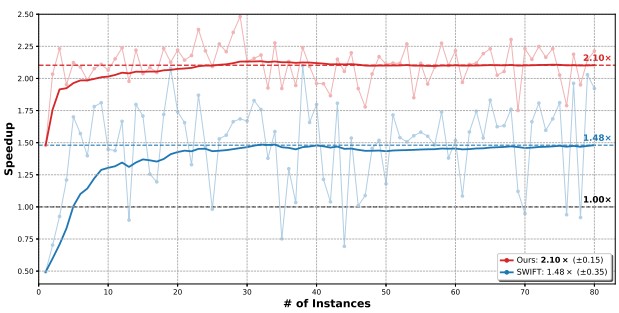

(a) Comparison with SWIFT on MT-Bench.

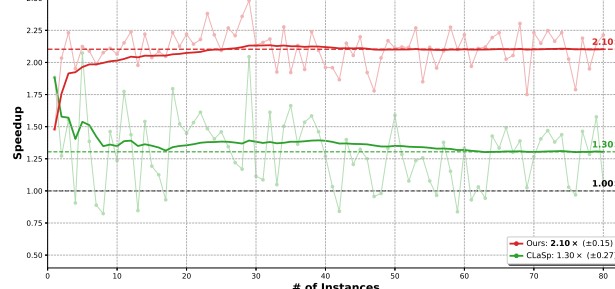

(b) Comparison with CLaSp on MT-Bench.

*Figure 5.* **Speedup comparison on MT-Bench.** We visualize the speedup of each instance (scattered points) and the average speedup (solid lines) across 80 random instances. (a) LEAP vs. SWIFT. (b) LEAP vs. CLaSp. LEAP demonstrates consistently higher speedup and efficiency compared to others.

SWIFT exhibits low initial speedup and gradually increases over multiple instances, resulting in poor early-stage efficiency. CLaSP maintains a relatively low average speedup across instances. Our efficiency primarily stems from zone partitioning and layer grouping, where the former quickly guides MCTS toward high-speedup configurations initially and the latter significantly reduces the search space, jointly enabling efficient exploration under a limited search budget. However, SWIFT initializes with random configurations and optimizes over multiple preceding instances, resulting in low efficiency at the beginning. CLaSP does not suffer from a cold-start issue, as it adaptively explores layer configurations for each instance. However, its exploration is guided by single-token feature alignment rather than direct inference speedup, which limits its ability to achieve substantial acceleration although without delay.

Efficiency of LEAP under different maximum generation lengths and different number of search iterations are respectively specified in Appendix D.1 and D.2.

**Performance.** Our method consistently maintains higher speedup across all instances, achieving an average speedup of $2.10\times$, whereas SWIFT and CLaSp achieve average speedup of approximately $1.48\times$ and $1.30\times$, respectively, both with greater variance. The superiority of LEAP primarily stems from the MCTS which efficiently explores layer configurations based on real-time inference speedup feedback and incrementally constructs the effective draft model for speculative decoding. In contrast, the other methods face the challenges of search capability and efficiency, which limits their ability to effectively identify potential configurations. As a result, their speedup remains lower and fluctuate more across instances.

### 5.4.2. LATENCY BREAKDOWN

Table 3 shows the per-instance average latency breakdown in LEAP using LLaMA-3-8B, including redundancy met-

*Table 3.* Per-instance latency of different components.

| Components | Latency (ms) | Ratio (%) |
|---|---|---|
| Metric Calculation | 12.31 | 0.066 |
| Zone Partitioning | 0.78 | 0.004 |
| Layer Grouping | 0.07 | 0.0004 |
| MCTS | 5.37 | 0.029 |
| *Subtotal (extra four)* | 18.53 | 0.0994 |
| Draft | 13127.62 | 69.8 |
| Verify | 5485.71 | 29.2 |
| Others | 175.02 | 0.9006 |
| Total | 18806.89 | 100.000 |

ric calculation, zone partitioning, layer grouping, MCTS, draft, verification and others. The results show that the first four newly introduced components totally incur negligible latency compared to the overall inference time. Specifically, redundancy metric calculation, zone partitioning, and layer grouping together account for less than 0.1% of the total latency, as they are executed only once in the whole inference and incur lightweight calculation. The MCTS contributes only 0.03% of the total latency which is insignificant compared to draft and verification, because it is conducted online solely during the early decoding stage.

The results demonstrate that although our method introduces several new components, they incur negligible overhead but significantly contribute to identifying potential layer configurations, thereby leading to higher final speedup.

### 5.4.3. EVALUATION ON DYNAMIC DATA STREAMS

Table 4 evaluates the robustness of different self-speculative decoding methods under dynamic input streams. We consider two stream settings: (i) *Sequential*, where instances from multiple datasets are arranged in a fixed dataset order, and (ii) *Random*, where instances are randomly shuffled across datasets. Each stream is constructed by sampling 40 instances from each of GSM8K, MT-Bench, CNN/DM, NQ, and WMT14 DE-EN. As shown in Table 4, LEAP achieves

*Table 4.* Speedup comparison between different methods with dynamic data input streams.

| Methods | Sequential | | Random | | Overall |
| --- | --- | --- | --- | --- | --- |
| | $M$ | Speedup | $M$ | Speedup | **Speedup** |
| LEAP | 4.65 | **2.03**× | 4.62 | **1.98**× | **2.01**× |
| SWIFT | 2.40 | 0.93× | 4.11 | 1.15× | 1.04× |
| CLaSP | 2.37 | 1.42× | 2.37 | 1.44× | 1.43× |

consistently high speedups under both settings, achieving $2.03\times$ on *Sequential* and $1.98\times$ on *Random*. This robustness comes from its instance-specific MCTS optimization, which adapts the draft configuration to each input rather than relying on a fixed configuration across different input distributions.

In contrast, SWIFT is sensitive to dynamic input streams and suffers from substantial speedup degradation. Since SWIFT uses the first few instances to guide Bayesian optimization and then applies the selected configuration to the remaining instances, the resulting configuration may not generalize well for different input types. CLaSP performs per-instance optimization and therefore remains relatively stable, but its speedup is moderate, suggesting that its search strategy is less effective at identifying high-speedup draft configurations.

Overall, these results show that LEAP adapts effectively to dynamical inputs, making it better suited for practical deployment scenarios where requests may come from diverse tasks and distributions.

## 6. Discussions

**Broader positioning.** Our main experiments compare training-free self-speculative decoding methods to fairly evaluate draft model construction strategies under the same constraints. Beyond this scope, Appendix E positions LEAP against broader approaches, including training-free methods outside the self-speculative decoding paradigm and training-based methods such as EAGLE-3. These comparisons show that LEAP targets a different practical trade-off: plug-and-play use and online adaptation, whereas training-based methods may achieve higher speedup at the cost of target-specific training.

**Broader settings.** Our main experiments use batch size 1 on relatively deep LLaMA-3 models with sufficient layer redundancy. To examine LEAP beyond this default setting, Appendix F further evaluates larger batch sizes, long-context inputs, and a smaller model LLaMA-3.2-1B. These results provide a broader evaluation of LEAP and characterize its generality across practical deployment settings.

**Limitations.** LEAP relies on empirical observations about prefilling-derived redundancy and zone-wise layer behavior, which may not hold equally well in all settings. For example, when the zone structure becomes coarse, the structured search space may provide a weaker inductive bias for MCTS, leading to lower acceleration. However, in our preliminary analysis and main experiments, these observations are stable enough to provide useful structural priors. Importantly, redundancy metrics and zone partitioning only initialize and constrain the search space rather than directly determine the final speedup. The final configuration is still selected online by MCTS with real-time speedup feedback, allowing LEAP to adapt dynamically.

In addition, the main limitation is usually not that MCTS fails to find a good configuration. Rather, LEAP becomes less beneficial when the optimization space is limited or the zone structure is less informative, yielding only modest speedup. This can happen in practical settings: smaller models such as LLaMA-3.2-1B have weaker redundancy and coarser zone partitioning; long-context inputs introduce additional overhead; and larger batch sizes suffer from batched inference and heterogeneous acceptance behaviours. Detailed results and analyses are provided in Appendix F.

## 7. Conclusion

In this work, we introduce LEAP, a plug-and-play self-speculative decoding approach that organizes layers into zones and groups based on prefilling-derived redundancy information, and leverages MCTS to explore potential layer configurations effectively within the structured search space. Extensive experiments demonstrate its superiority on both performance and efficiency.

## Acknowledgements

This work was supported in part by the Nature and Science Funding of Guangdong Province under Grant 2026A1515011230 and in part by the Science and Technology Program of Shenzhen under Grant ZDCY20250901112705007.

## Impact Statement

This paper presents work whose goal is to advance the field of Machine Learning. There are many potential societal consequences of our work, none which we feel must be specifically highlighted here.

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

# A. Pseudo Code for LEAP

Here we give all algorithms of our LEAP in detail. Algorithm 1 illustrates the initialization including metric calculation, zone partitioning and layer grouping during the prefilling stage. Algorithm 2 represents standalone MCTS with selection, expansion, simulation and backpropagation. Algorithm 3 illustrates the speculative decoding iterations with MCTS.

---

**Algorithm 1** Prefilling-Based Initialization

---

**Input:** input tokens $\mathbf{x} = \{x_1, \ldots, x_N\}$, target LLM $M_p$ with $M$ layers, extract window length $L$, percentiles $\rho_1, \rho_2$
**Output:** layer groups $\mathcal{G} = \{G_1, \ldots, G_m\}$ with $m$ groups
**Stage 1: Feature and Probability Extraction**
Initialize storage $\mathcal{H} \leftarrow \emptyset, \mathcal{P} \leftarrow \emptyset$
Let $\mathbf{W}_{head}$ be the weight matrix of the shared LM Head
Completed in one prefilling forward pass
**for** $l = 1$ **to** $M$ **do**
    Standard forward pass for layer $l$
    $\mathbf{h}^{(l)} \leftarrow \text{TransformerBlock}_l(\mathbf{h}^{(l-1)})$
    Append $\mathbf{h}^{(l)}[N - L : N]$ to $\mathcal{H}$
    $\mathbf{q}^{(l)} \leftarrow \text{Softmax}(\text{LayerNorm}(\mathbf{h}^{(l)}[N - L : N]) \cdot \mathbf{W}_{head}^T)$
    Greedy sampling
    $\mathbf{p}^{(l)} \leftarrow \max(\mathbf{q}^{(l)}, \dim = -1)$
    Append $\mathbf{p}^{(l)}$ to $\mathcal{P}$
**end for**
**Stage 2: Metric Calculation**
Initialize metric lists $\mathcal{R} \leftarrow [], \mathcal{D} \leftarrow [], \bar{L}_{prev} \leftarrow 0$
$\mathbf{p}_{target} \leftarrow \mathcal{P}[M]$
**for** $l = 1$ **to** $M$ **do**
    Calculate $R_l$
    $R_l \leftarrow \|\mathcal{H}[l] - \mathcal{H}[l-1]\|_F / (\|\mathcal{H}[l-1]\|_F)$
    Append $R_l$ to $\mathcal{R}$
    Calculate $\Delta \bar{L}_l$
    $\boldsymbol{\alpha} \leftarrow \min(1.0, \mathbf{p}_{target} / (\mathcal{P}[l] + \epsilon))$
    $\bar{L}_{curr} \leftarrow \text{Sum}(\text{CumProd}(\boldsymbol{\alpha}))$
    $\Delta \bar{L}_l \leftarrow \bar{L}_{curr} - \bar{L}_{prev}$
    Append $\Delta \bar{L}_l$ to $\mathcal{D}$
    $\bar{L}_{prev} \leftarrow \bar{L}_{curr}$
**end for**
**Stage 3: Zone Partitioning & Grouping**
Calculate thresholds: $\tau_R \leftarrow \text{Percentile}(\mathcal{R}, \rho_1), \tau_L \leftarrow \text{Percentile}(\mathcal{D}, \rho_2)$
Initialize Zone map $\mathcal{Z} \leftarrow \emptyset$
**for** $l = 1$ **to** $M$ **do**
    **if** $R_l > \tau_R$ **then**
        $\mathcal{Z}[l] \leftarrow \text{Early}$
    **else if** $\Delta \bar{L}_l > \tau_L$ **then**
        $\mathcal{Z}[l] \leftarrow \text{Final}$
    **else**
        $\mathcal{Z}[l] \leftarrow \text{Middle}$
    **end if**
**end for**
$\mathcal{G} \leftarrow \text{GroupingStrategy}(\mathcal{Z})$
**return** $\mathcal{G}$

---

---

**Algorithm 2** Zone-Structured MCTS for Draft Model Construction

---

**Input:** layer groups $\mathcal{G}$, zone map $\mathcal{Z}$, max MCTS iterations $K$, exploration coefficient $c$, depth penalty coefficient $\lambda$
Initialize Root $v_0$ (empty configuration)
**for** $k = 1$ **to** $K$ **do**
    **1. Selection**
    $v_{curr} \leftarrow v_0$
    **while** $v_{curr}$ is not leaf **and** $v_{curr}$ is fully expanded **do**
        $v_{curr} \leftarrow \text{argmax}_{v' \in \text{children}(v_{curr})} \text{UCB}(v', v_{curr}, c, \lambda)$
    **end while**
    **2. Expansion (Zone-Constrained)**
    **if** $v_{curr}$ is not terminal **then**
        $G_{next} \leftarrow \text{NextGroup}(v_{curr})$
        $\mathcal{A}_{valid} \leftarrow \text{GetZoneActions}(G_{next}, \mathcal{Z})$
        $v_{child} \leftarrow \text{Expand}(v_{curr}, \text{Unexplored}(\mathcal{A}_{valid}))$
    **else**
        $v_{child} \leftarrow v_{curr}$
    **end if**
    **3. Simulation (via Speculative Decoding)**
    $C_{partial} \leftarrow \text{GetConfig}(v_{child})$
    Select actions for remaining groups via rollout policy
    $C_{complete} \leftarrow \text{Rollout}(C_{partial}, \mathcal{G})$
    Run speculative decoding to get real-time speedup feedback and reward
    $R \leftarrow \textbf{SpeculativeDecoding}(C_{complete})$
    **4. Backpropagation**
    $\text{Backpropagate}(v_{child}, R)$
**end for**
**return** $\text{BestChild}(v_0)$

---

**Algorithm 3** Speculative Decoding Iterations with MCTS

---

**Input:** input $\mathbf{x}$, target model $\mathcal{M}_p$, token tree length $L$, token tree width $W$, max MCTS iterations $K$
**Output:** generated sequence $\mathbf{y}$
**Initialization**
$\mathcal{G}, \mathcal{Z} \leftarrow \text{Initialize}(\mathbf{x}, \mathcal{M}_p)$
Initialize MCTS State $S_{tree}$
$\mathbf{y} \leftarrow \mathbf{x}$
**for** $k = 1$ **to** $K$ **do**
    Detailed in Algorithm 2 to identify optimal layer configuration
    $C^* \leftarrow \text{MCTS}(\mathcal{G}, \mathcal{Z}, S_{tree})$
    $\mathcal{M}_D \leftarrow \text{ConstructDraftModel}(\mathcal{M}_p, C^*)$
    Initialize Token Tree $\mathcal{T} \leftarrow \mathbf{y}$
    $\mathcal{T} \leftarrow \text{TreeGeneration}(\mathcal{M}_D, L, W)$
    $\mathbf{P}_{verify} \leftarrow \mathcal{M}_p(\mathcal{T})$
    $\mathbf{y}_{new}, \text{acc\_len} \leftarrow \text{TreeVerify}(\mathcal{T}, \mathbf{P}_{verify})$
    $\mathbf{y} \leftarrow \text{Concat}(\mathbf{y}, \mathbf{y}_{new})$
    Speedup feedback for MCTS
    $Speedup \leftarrow \text{CalculateSpeedup}(\text{acc\_len}, \text{Time}_{\text{draft}}, \text{Time}_{\text{verify}})$
    $R \leftarrow \text{CalculateReward}(Speedup)$
    $\text{UpdateMCTS}(S_{tree}, C^*, R)$
**end for**
**return** $\mathbf{y}$

---

## B. Experimental Setups

### B.1. Dataset Configurations

Our experiments mainly evaluate the inference acceleration of LEAP on different categories of tasks including mathematical reasoning, multi-round dialogue, abstractive summarization, open-domain question answering and machine translation. Specifically, we choose GSM8K (Cobbe et al., 2021) which consists of grade-school math problems to evaluate multi-step numerical reasoning, MT-Bench (Zheng et al., 2023) which includes diverse dialogue questions to evaluate the conversational ability, CNN/DM (Nallapati et al., 2016) which contains news articles and requires summaries to evaluate abstractive summarization, Natural Question (Kwiatkowski et al., 2019) which consists of real-world questions requiring reasoning and open-domain question answering, and WMT14 DE-EN which comprises parallel German–English sentence pairs to evaluate machine translation quality.

### B.2. Model Configurations

We conduct our experiments on LLaMA3 series (Grattafiori et al., 2024) choosing LLaMA-3-8B and LLaMA-3-70B . In our experiments, all models are loaded in the precision of float-16. Our LEAP directly uses these models with no need of any additional module, training and fine-tuning.

### B.3. Inference Setups

Models are deployed on NVIDIA H20 GPUs with 96GB of memory. Specifically, LLaMA-3-8B is deployed on a single H20 GPU for all methods, and LLaMA-3-70B is deployed on two H20 GPUs. For inference, we use a batch size of 1, which is more consistent with actual deployment and is commonly used in other speculative decoding methods. When comparing with other self-speculative decoding methods, we use the same model configuration and device usage for fairness.

For the initialization in LEAP, we set the zone-partitioning thresholds for $R_l$ and $\Delta \bar{L}$ as the 82.5$^{\text{th}}$ percentiles, respectively. For MCTS, we set the exploration coefficient $c$ as 4.0 and the depth penalty coefficient $\lambda$ as 0.3. To trade off the search efficiency and performance of MCTS while adapting to different generation lengths, we set the maximum MCTS iterations $N_{\text{search}}$ dynamically based on the number of layer groups $|\mathcal{G}|$:

$$N_{\text{search}} = \max \left( 15, \min \left( 2 \cdot |\mathcal{G}|, 40 \right) \right). \tag{6}$$

MCTS can also be terminated if the optimal configuration remains unchanged for 5 consecutive iterations. At last we set the length of the draft token tree $L$ as 4 and the width $W$ as 8.

For SWIFT, we set the hyperparameters according to its original settings. We set the context window $\gamma$ as 50, the interval of Bayesian optimization $\beta$ as 25, the maximum iterations of optimization $S$ as 1000, the maximum draft length $N_D$ as 25, and the skip ratio $r$ as 0.45 for LLaMA-3-8B and 0.5 for LLaMA-3-70B. The optimization is set to be early stopped if the matchness score does not improve after 300 steps or exceeds 0.93. For CLaSP, we set the skip ratio $r$ as 0.5 for two models and the interval of layer optimization $\beta$ as 64.

## C. Details of Grouping Procedure

After zone partitioning, LEAP further merges consecutive layers into groups, which serve as the basic decision units for MCTS. The grouping strategy is deterministic and zone-aware. It is based on the zone assignment together with the redundancy signals $R_\ell$ and $\Delta \bar{L}$, and consists of two steps: determining adaptive group size and merging layers within each zone into groups.

The group size is not fixed globally. Instead, LEAP uses zone-adaptive group size. Let $n_2$ and $n_3$ denote the numbers of layers assigned to the middle zone and final zone, respectively. The group sizes are set as

$$g_2 = \max \left( 3, \left\lfloor \frac{n_2}{6} \right\rfloor \right), \tag{7}$$

$$g_3 = \max \left( 2, \left\lfloor \frac{n_3}{4} \right\rfloor \right). \tag{8}$$

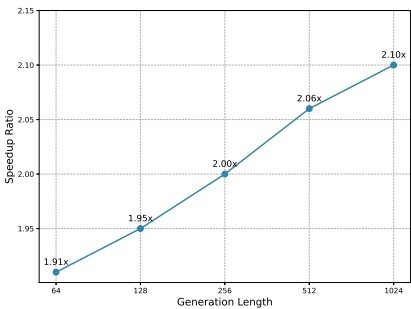

*Figure 6.* Speedup ratio across different generation lengths.

Therefore, when a zone contains more layers, slightly larger groups are allowed; when it is shorter, the groups remain compact. This adaptive design enables the search space to adapt to the size and structure of each zone.

For merging, LEAP traverses layers from shallow to deep. Crossing a zone boundary always triggers a split, so each group contains layers from only one zone. In the early zone, all consecutive layers are merged into a single group because they share the same action `execute`. In the middle zone, LEAP identifies sensitive layers according to large $\Delta \bar{L}$ or sharp increases in $R_l$. Transitions between sensitive and non-sensitive layers trigger a split, while consecutive non-sensitive layers are merged until the group size is reached.

In the final zone, LEAP adopts a more conservative grouping strategy because these layers are closer to the target prediction. Layers with high or sharply increased $\Delta \bar{L}$ trigger new groups, while less sensitive consecutive layers are merged until the group size is reached. This protects layers with strong effects on acceptance length while reducing the MCTS search space.

## D. Additional In-Depth Analyses

### D.1. Efficiency of LEAP across Generation Lengths

In the main experiments, we set the maximum generation length as 1024 and the maximum draft length as 4. As a result, LEAP can leverage a sufficient number of iterations to effectively perform MCTS and identify potential configurations for acceleration. To further evaluate the efficiency of LEAP, we measure its average speedup on GSM8K, MT-Bench and WMT14 DE-EN, under maximum generation lengths of $\{64, 128, 256, 512, 1024\}$.

As shown in Figure 6, the speedup ratio increases from $1.91\times$ to $2.10\times$ as the maximum generation length increases from 64 to 1024. This trend indicates that longer generation lengths require more speculative decoding iterations to produce the output, enabling a sufficient number of MCTS iterations to progressively refine effective layer configurations.

Notably, when the maximum generation length reduces to 64, LEAP can still achieve a substantial speedup of $1.91\times$, with marginal degradation compared to those of larger generation lengths. Under the short generation length, the speculative decoding process often terminates before MCTS completes its search to construct the effective draft model. However, the resulting speedup degradation remains marginal, directly reflecting the efficiency and effectiveness of the MCTS-based optimization. By leveraging redundancy-aware zone partitioning and layer grouping, which jointly construct the structured search space and initially avoid low-return layer configurations for MCTS, LEAP can rapidly identify and focus on potential configurations during MCTS.

### D.2. Ablation Study on the Number of MCTS Iterations

To further evaluate the efficiency of LEAP, we conduct ablation studies on the number of MCTS iterations. In the main experiments, we dynamically determine it by the number of layer groups as defined in Equation 6, which is generally in the range of $25 \sim 35$, and we further examine the speedup when the number of MCTS iterations is $\{10, 20, 40, 60, 80\}$. As shown in Table 5, as the number of iterations increases from 10 to 80, LEAP gradually achieves higher speedup, indicating that a larger search budget enables MCTS to more effectively explore potential layer configurations.

Notably, the marginal speedup gradually diminishes as the number of iterations increases, suggesting that the search converges to sufficiently high-speedup configurations after a moderate number of iterations. This trend demonstrates the

*Table 5.* Ablation on the number of MCTS search iterations across different datasets.

| MCTS Iterations | GSM8K | MT-Bench | WMT14 |
|---|---|---|---|
| 10 | 1.59× | 1.73× | 2.05× |
| 20 | 1.77× | 1.83× | 1.90× |
| 40 | 1.92× | 1.96× | 2.01× |
| 60 | 2.08× | 1.94× | 2.20× |
| 80 | **2.18×** | 1.95× | **2.25×** |
| LEAP | 2.12× | **2.07×** | 2.11× |

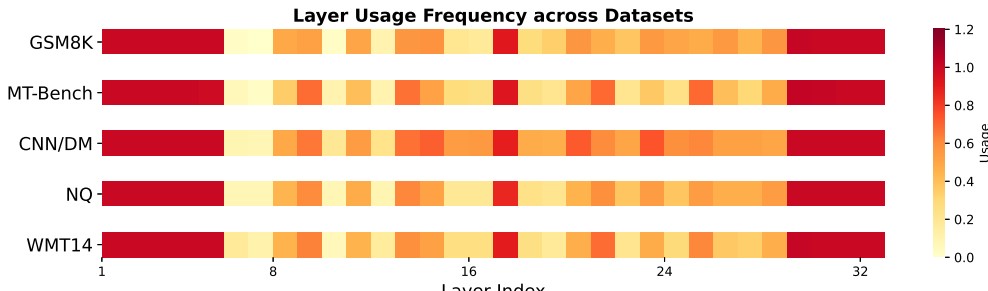

*Figure 7.* Per-layer usage frequency across datasets.

feasibility of a dynamic number of MCTS iterations. As shown in the last row in Table 5, LEAP with dynamic number of MCTS iterations achieves competitive or superior speedup compared to fixed ones. This demonstrates that LEAP can adaptively determine its number of MCTS iterations based on the size of search space, effectively balancing the search efficiency and the final speedup.

In summary, while increasing the MCTS search budget generally achieves higher speedup, LEAP does not critically require a large and fixed number of MCTS iterations. Instead, its redundancy-aware zone partitioning and layer grouping enable MCTS to focus on promising configurations rapidly, enabling robust and efficient acceleration under a relatively small and dynamic search budget.

### D.3. Per-Layer Usage Frequency

Figure 7 illustrates the per-layer usage frequency across datasets. For the early zone, the first few layers are consistently executed with usage frequency of 1, since they are important for preserving stable feature transformation. In the middle zone, the usage frequency is generally lower and varies across layers, indicating that MCTS tends to skip redundant layers while still preserves some useful layers. For the final zone, the usage frequency remains close to or slightly above 1, showing that `repeat` is only selected occasionally as an optional compensation for aggressive skipping rather than a necessary operation

Across datasets, the average retained-layer ratio of LEAP is about 56%. This ratio is viewed as the result of the speedup trade-off rather than an objective to be minimized. Reducing the retained-layer ratio can lower draft computation, but may also degrade draft quality and decrease the accepted length, which can reduce the final speedup. Therefore, LEAP does not simply minimize the number of retained layers; instead, it uses online MCTS to adaptively balance the draft cost and quality.

### D.4. Draft Token Tree Length and Width

Figure 8 illustrates the impact of token tree size on speedup, including both the token tree length and width. As shown in Figure 8a, increasing the token tree length from 3 to 4 consistently improves speedup and achieves the highest one in the range of $3 \sim 8$, indicating that moderately deeper token tree enables LEAP to explore more candidate tokens within one iteration. However, as the tree goes deeper, the token-level alignment between the draft and target models gradually decreases. Consequently, further increases in tree length incur additional draft overhead without improving the generation length, ultimately reducing the overall speedup.

Figure 8b shows a similar trend with respect to token tree width. Increasing the tree width gradually improves the speedup

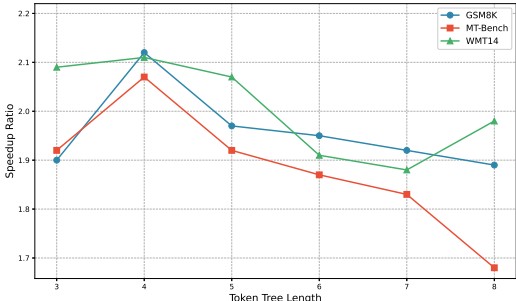

*(a)* Speedup across different token tree lengths.

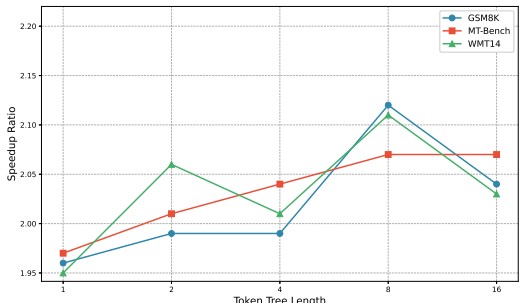

*(b)* Speedup across different token tree widths.

*Figure 8.* Effect of token tree structure on speedup across three datasets.

*Table 6.* Comparison with training-free speculative decoding methods beyond the self-speculative setting.

| Dataset | LEAP | MTAD | DSBD | Lookahead |
|---------|------|------|------|-----------|
| GSM8K | **2.02**× | 1.95× | 1.58× | 1.87× |
| MT-Bench | 2.03× | **2.34**× | 1.59× | 1.85× |

to a relatively high level, after which it exhibits a slight decline. A larger tree width implies more candidate tokens at the same position, which increases the acceptance rate at that position, but it also incurs higher draft and verification overhead, highlighting a trade-off between the improved acceptance rate and increased overhead.

Overall, these results indicate that both token tree length and width should be carefully configured to achieve effective acceleration. Excessively increasing them can not consistently improve speedup. Accordingly, a tree length of 4 and width of 8 as used in the main experiments constitute an appropriate configuration to achieve effective speedup without incurring additional overhead.

## E. Broader Comparison with Speculative Decoding Methods

Our main experiments focus on training-free self-speculative decoding methods, since LEAP is designed to optimize draft model construction within the self-speculative paradigm. To better position LEAP in the broader speculative decoding literature, we provide additional comparisons with two categories of methods: training-free methods outside self-speculative decoding, and training-based method EAGLE-3. These comparisons are intended to clarify the practical trade-offs between speedup, training cost, and deployment simplicity.

### E.1. Comparison with Training-Free Non-Self-Speculative Methods

We first compare LEAP with training-free speculative decoding methods outside the self-speculative paradigm, including MTAD (Qin et al., 2025b), DSBD (Qin et al., 2025a), and Lookahead (Fu et al., 2024). Unlike LEAP, these methods do not construct the draft model by selecting layers from the target model, but instead use alternative drafting mechanisms such as a smaller model from the same model family or token-level lookahead. For a fair comparison, we use the same target model

*Table 7.* Comparison with EAGLE-3.

| Method | Training free | GSM8K | CNN/DM | WMT14 | NQ | MT-Bench | Overall |
|--------|---------------|-------|--------|-------|-----|----------|---------|
| LEAP | Yes | 1.89× | 1.71× | 1.87× | 1.89× | **1.84×** | 1.84× |
| EAGLE-3 | No | **2.99×** | **2.48×** | **2.50×** | **2.59×** | 1.75× | **2.44×** |

*Table 8.* Speedup under different batch sizes.

| Batch size | 1 | 2 | 4 | 8 |
|------------|-----|-----|-----|-----|
| LEAP | **2.02×** | **1.87×** | **1.60×** | **1.41×** |
| CLaSP | 1.47× | 1.49× | 1.36× | 1.13× |

LLaMA-3-8B at Temperature=0. For MTAD and DSBD, we use LLaMA-3.2-1B as the draft model.

As shown in Table 6, LEAP achieves the best speedup on GSM8K. On MT-Bench, LEAP outperforms DSBD and Lookahead, while MTAD achieves higher speedup, partly because it adopts lossy verification, which improves acceptance and latency but does not strictly preserve the target probability distribution. In contrast, LEAP follows lossless verification, ensuring that the final output distribution remains identical to that of the target model.

Overall, LEAP remains highly competitive among training-free non-self-speculative decoding methods. More importantly, it provides plug-and-play acceleration once the target model is available, reflecting a practical trade-off between speedup and deployment simplicity.

### E.2. Comparison with EAGLE-3

We further compare LEAP with the training-based method EAGLE-3 using LLaMA-3.1-8B-Instruct as the target model. Unlike LEAP, EAGLE-3 uses an additional trained drafter to improve draft-target alignment, so this comparison mainly clarifies the trade-off between absolute speedup and deployment simplicity. As shown in Table 7, EAGLE-3 achieves higher overall speedup than LEAP, which is expected because of its extra optimization capacity beyond self-speculative decoding. However, this gap reflects a fundamental difference in practical objectives rather than strict superiority.

Specifically, EAGLE-3 pursues stronger acceleration at the cost of target-specific drafter training. Although EAGLE provides checkpoints for many target models, its coverage still remains relatively limited, and unsupported target models require users to train a specific drafter with substantial data and computational overhead. In contrast, LEAP requires neither an auxiliary drafter nor target-specific training, and can be applied immediately once the target model is available. This makes LEAP suitable for scenarios that require a training-free, plug-and-play SD method under limited computational resources.

Overall, EAGLE-3 and LEAP reflect different practical trade-offs: EAGLE-3 emphasizes higher speedup with extra training, while LEAP emphasizes training-free usability and deployment simplicity.

## F. Evaluation across Broader Deployment Settings

The main experiments evaluate LEAP with a batch size of 1 and relatively deep LLaMA-3 models. To further evaluate the applicability and limitations of LEAP, we conduct additional evaluations under broader deployment settings, including larger batch sizes, long-context inputs and a smaller model. These settings help characterize when LEAP remains effective and when its acceleration becomes more limited.

### F.1. Efficiency under Different Batch Sizes

Although speculative decoding is commonly evaluated with batch size 1, modern LLM deployment increasingly relies on batched inference for higher serving throughput. Therefore, we further evaluate LEAP under different batch sizes to evaluate its efficiency in batched inference scenarios.

As shown in Table 8, LEAP remains beneficial under batched inference and consistently outperforms CLaSP across larger batch sizes. The speedup of both methods decreases as batch size increases, which is expected because batched inference makes drafting and verification heavier, while heterogeneous token acceptance behaviors of samples in the same batch reduce the benefit of speculation.

*Table 9.* Speedup of LEAP under different input lengths.

| Dataset | 8K | 32K | 64K | 128K |
|---|---|---|---|---|
| LongBench-v2 | 1.87× | 1.77× | 1.85× | 1.85× |

*Table 10.* Speedup of LEAP on the smaller Llama-3.2-1B model.

| Method | GSM8K | MT-Bench | CNNDM |
|---|---|---|---|
| LEAP | **1.58×** | **1.70×** | **1.52×** |
| CLaSP | 1.33× | 1.26× | 1.18× |

LEAP is designed as an algorithmic self-speculative decoding method rather than a specialized batch-serving optimization framework, but it still maintains positive acceleration under batched inference, showing its efficiency beyond the batch-size-1 setting.

### F.2. Efficiency under Long-Context Inputs

We further evaluate LEAP on long-context inputs using LLaMA-3-8B on LongBench-v2 (Bai et al., 2025), with input lengths of {8K, 32K, 64K, 128K}.

As shown in Table 9, LEAP achieves consistent speedups from 1.77× to 1.87× across long-context input lengths. The gains are slightly lower than those in the main experiments, indicating that long-context inference is a more challenging regime due to heavier prefilling overhead and larger KV cache. Nevertheless, LEAP still provides clear acceleration under long-context settings, indicating that it remains beneficial in this regime.

### F.3. Efficiency under Smaller Model

We also evaluate LEAP on Llama-3.2-1B, a smaller model with 16 layers. This setting examines whether LEAP remains effective when the model provides a smaller search space for layer-configuration optimization. As shown in Table 10, LEAP achieves speedups of 1.58×, 1.70×, and 1.52× on GSM8K, MT-Bench and CNN/DM, respectively, consistently outperforming CLaSP. This indicates that LEAP remains effective in the small-model regime.

Nevertheless, the speedup is lower than that on larger models. The degradation is expected because smaller models have fewer layers and weaker layer redundancy, which limits the optimization space. In addition, the zone partitioning becomes coarser, making the structural prior less informative for MCTS. This result is consistent with our discussion that LEAP provides smaller gains when the zone structure is less informative or the exploitable redundancy is limited.

### F.4. Summary of Broader Settings

Overall, these additional evaluations show that LEAP remains effective beyond the main experimental setting, but its gain is affected by the deployment setting and model scale. Specifically, its speedup becomes more moderate under larger batch sizes and long contexts which both introduce additional overhead, or smaller models with limited layer redundancy.

These results support the discussion in Section 6: LEAP's limitation is better viewed as low-gain regimes rather than failure cases. When the optimization space is limited or the zone structure is less informative, LEAP may yield more modest speedup, but it still remains applicable without additional modules or training.

