# OpenReview forum: "LEAP: Zone-Aware MCTS for LLM Self-Speculative Decoding"
_ICML.cc/2026/Conference — ICML 2026 regular_

### Official Review · Reviewer_oDii · 2026-02-18

**Soundness:** 3
**Presentation:** 3
**Significance:** 3
**Originality:** 3
**Overall Recommendation:** 4
**Confidence:** 4

**Summary:**

This paper proposes LEAP, a plug-and-play self-speculative decoding framework that uses Monte Carlo Tree Search (MCTS) to optimize which layers of an LLM to construct a lightweight draft model. The key idea is to partition layers into three zones (early, middle, final) based on redundancy metrics computed during prefilling, then use MCTS to search over layer-group configurations during decoding. The method achieves 1.7×–2.0× speedup on LLaMA-3-8B and 70B across several benchmarks.

**Compliance With Llm Reviewing Policy:**

Affirmed.

**Final Justification:**

The author addressed my concerns in their second response, I am open to accepting it.

**Key Questions For Authors:**

How does the zone partitioning change across different input prompts for the same model? Is it stable enough that you could precompute zones offline and skip the per-instance prefilling analysis?

What happens when the model has fewer layers (e.g., a 7B model with 32 layers vs. a 1B model with ~16 layers)? Does the zone structure still emerge meaningfully?

Could you provide wall-clock times (not just speedup ratios) so readers can understand the absolute latency improvements?
How does LEAP interact with other inference optimizations like quantization or KV cache compression?

**Limitations:**

Batch size 1 only is the main limiation from my perspective.

**Strengths And Weaknesses:**

Strengths

Good observation. The prefilling redundancyand decoding-time redundancy exhibits zone-wise structure are clearly presented.  I like the Spearman correlation analysis (Figure 1) and the zone visualization (Figures 2–3).

I like the experiments and ablation study. They clearly show LEAP substantially outperforms SWIFT and CLaSP across all five datasets, and the contribution of each component, and the additional experiments on generation length, MCTS iterations, token tree structure, and per-layer usage frequency provide useful insights.


Weakness

No analysis of failure cases or limitations. When does LEAP fail to find good configurations? Are there task types or input patterns where the zone partitioning is misleading? The paper presents a uniformly positive picture, which reduces its informativeness.

Batch size 1 only. All experiments use batch size 1. This is common in speculative decoding wok, but modern deployment increasingly involves batched inference.

---

> ### Author Rebuttal · Authors · 2026-03-31
>
> **Note.** All experiments are conducted with **Temperature=0**.
>
> ---
>
> **[Q1]: No analysis of failure cases or limitations. When does LEAP fail to find good configurations? Are there task types or input patterns where the zone partitioning is misleading?**
>
> **[A1]**: We thank the reviewer for this question. We agree that the manuscript emphasizes positive results and doesn't clearly explain when LEAP becomes less effective. The main limitation is usually **not** that MCTS fails to find a good configuration. Rather, LEAP becomes less beneficial when the **optimization space is limited** or the **zone structure is less informative**, so the final configuration yields only modest speedup.
>
> To clarify these boundary conditions, we add three analyses. First, on **smaller models** such as Llama-3.2-1B, LEAP remains effective, but the speedup drops because weaker redundancy and inter-layer heterogeneity make the zone partitioning coarser. Second, on **long-context inputs**, LEAP still provides clear acceleration, but the gain is lower due to additional overhead. Third, the **hyperparameter study** shows only moderate degradation, suggesting that these lower-gain cases mainly come from the model/task itself rather than hyperparameter sensitivity. We refer the reviewer to our response **Q2 to Reviewer LySS** for detailed results due to space constraint.
>
> Overall, LEAP’s limitation is better viewed as a **low-gain regime** rather than search failure: when the zone structure becomes coarser or the overhead becomes heavier, LEAP still provides meaningful acceleration, though the gain becomes smaller.
>
> ---
>
> **[Q2]: Batch size 1 only.**
>
> **[A2]**: We thank the reviewer for highlighting the importance of evaluating LEAP beyond batch-size-1. We refer the reviewer to our response **Q4 to Reviewer xsr8** for a detailed discussion. Overall, LEAP remains beneficial beyond BS=1 and still outperforms CLaSP, though its speedup drops with larger batches due to higher batched overhead and no dedicated batch-serving optimization.
>
> ---
>
> **[Q3]: How does the zone partitioning change across different prompts? Is it stable enough to precompute zones offline?**
>
> **[A3]**: We thank the reviewer for this insightful question. We address it from two aspects.
>
> For **variation of partitioning**, we find it considerably stable across prompts for the same model. As shown in **Figure 3 of manuscript**, although the zone boundaries may shift slightly, the overall pattern remains consistent. This suggests that LEAP does not rely on highly unstable prompt-specific partitioning.
>
> For **stability for offline precomputation**, we evaluate LEAP using a fixed offline partitioning from **Figure 3**: layers 1–7 as Zone 1, 8–27 as Zone 2, and 28–32 as Zone 3.
>
> | Dataset  | Online | Offline |
> | :------: | :----: | :-----: |
> |  GSM8K   | 2.02×  |  1.90×  |
> | MT-Bench | 2.03×  |  1.91×  |
>
> The offline partitioning remains effective, but is consistently slightly worse than the online one. This indicates that the zone structure is fairly stable, but not fully universal: small dataset-specific shifts can still reduce the speedup. In addition, zone boundaries also need adjustment for models with different depths.
>
> ---
>
> **[Q4]: What happens when the model has fewer layers? Does the zone structure still emerge meaningfully?**
>
> **[A4]**: We thank the reviewer for this insightful question. We evaluate LEAP on **Llama-3.2-1B**. We refer the reviewer to our response **Q2 to Reviewer LySS** for detailed results. Overall, LEAP remains competitive, though its speedup is moderately reduced by weaker layer redundancy and model redundancy.
>
> Moreover, the zone structure on Llama-3.2-1B is shown below. Although the partitioning becomes coarser, the 1B model can still be partitioned into three zones based on $\Delta EAR$ and $R_l$.
>
> | Model        | Dataset | #Layers | Zone 1 | Zone 2 | Zone 3 |
> | ------------ | ------- | ------- | ------ | ------ | ------ |
> | Llama-3.2-1B | GSM8K   | 16      | 1-3    | 4-11   | 12-16  |
>
> ---
>
> **[Q5]: Could you provide wall-clock times so readers can understand the absolute latency improvements?**
>
> **[A5]**: We apologize for not making the results clearer in the manuscript. Following the reviewer’s suggestion, we report the **real throughput (tokens/s)** together with the corresponding **speedup** to help readers understand the absolute latency improvements.
>
> |  Method  | GSM8K Tokens/s | GSM8K Speedup | MT-Bench Tokens/s | MT-Bench Speedup |
> | :------: | :------------: | :-----------: | :---------------: | :--------------: |
> | Vanilla  |     20.959     |     1.00×     |      20.731       |      1.00×       |
> |  SWIFT   |     17.56      |     0.78×     |       31.03       |      1.50×       |
> |  CLaSP   |     30.08      |     1.43×     |       31.07       |      1.50×       |
> | **LEAP** |   **42.31**    |   **2.02×**   |     **42.17**     |    **2.03×**     |

---

> > ### Author Rebuttal · Reviewer_oDii · 2026-03-31
> >
> > We appreciate the authors' response. After following their suggestion to refer Reviewer xsr8 to our response Q4 regarding their comments, I have come to agree with Reviewer xsr8's perspective: the authors should directly address the comparison with EAGLE-3 rather than sidestepping it. Consequently, we have decided to maintain our original rating.

---

> > > ### Author Response · Authors · 2026-04-01
> > >
> > > **[Q1]: The authors should directly address the comparison with EAGLE-3 rather than sidestepping it.**
> > >
> > > **[A1]**: We thank the reviewer for this valuable suggestion. We agree that a direct comparison with EAGLE-3 is useful for broader positioning, and we will include it in the revised paper. We have now completed this comparison with **Llama-3.1-8B-Instruct**.
> > >
> > > | Method  | Training-free | GSM8K | CNN/DM | WMT14_DEEN |  NQ   | MT-Bench | Overall |
> > > | :-----: | :-----------: | :---: | :----: | :--------: | :---: | :------: | :-----: |
> > > |  LEAP   |      Yes      | 1.89× | 1.71×  |   1.87×    | 1.89× |  1.84×   |  1.84×  |
> > > | EAGLE-3 |      No       | 2.99× | 2.48×  |   2.50×    | 2.59× |  1.75×   |  2.44×  |
> > >
> > > From the table, we observe that **EAGLE-3 achieves higher absolute speedup** than LEAP. We believe this is expected, since EAGLE-3 relies on an additional **trained drafter**, which gives it additional optimization capacity compared with self-speculative decoding methods. **This performance gap, however, can not diminish the value of LEAP**, since LEAP targets a **different practical objective**: training-free, plug-and-play self-speculative decoding.
> > >
> > > In addition, we would like to clarify two points. **First**, LEAP and EAGLE-3 target different deployment trade-offs. While EAGLE-3 achieves stronger speedup, it requires additional training cost and time. LEAP instead aims at a **plug-and-play, training-free** solution that can be applied immediately once the target model is available. **Second**, the reason we do not include EAGLE-3 in the manuscript is that our main experiments specifically focus on isolating and comparing  **different ways of constructing the draft model within the self-speculative paradigm**, rather than comparing against SD methods with trained drafters or drafters in the same model family as the target model.
> > >
> > > Overall, we agree that EAGLE-3 is an **important stronger baseline** for broader comparison, and we expect that the added results help evaluate LEAP more clearly. At the same time, we believe this comparison also highlights the practical value of LEAP as a **training-free self-speculative approach**, offering a different trade-off between speedup, simplicity, and deployment cost.

---

### Official Review · Reviewer_1Ecd · 2026-03-10

**Soundness:** 3
**Presentation:** 2
**Significance:** 3
**Originality:** 3
**Overall Recommendation:** 5
**Confidence:** 2

**Summary:**

The paper proposes LEAP, a plug-and-play self-speculative decoding framework that constructs a draft model by selecting, skipping, or repeating groups of layers from the target LLM. LEAP frames draft construction as a sequential decision process and applies a zone-aware MCTS that uses prefilling-derived redundancy signals and zone/group structuring to constrain the search and directly optimize wall-time speedup during lossless speculative decoding. Empirically, LEAP reports consistent 1.7×–2.0× end-to-end speedups over vanilla decoding on LLaMA-3 8B/70B across several datasets, outperforming prior training-free self-speculative baselines.

**Compliance With Llm Reviewing Policy:**

Affirmed.

**Final Justification:**

The author answered my questions in detail and thoroughly explained the algorithm details and settings. I suggest that the author include these details in the article so that readers can understand how the system works. I will keep the current score and confidence.

**Key Questions For Authors:**

1. What is the exact definition and allowed multiplicity of the “repeat” action? How many repeats are permitted, and how frequently does LEAP select repeats in practice?
2. How is the grouping strategy determined (e.g., fixed group sizes, heuristic merging)? Please provide the algorithmic details.
3. How does LEAP scale under batch>1 and longer contexts?

**Limitations:**

yes

**Strengths And Weaknesses:**

Strengths:
1. Technical novelty and innovation:
  - Formulating layer configuration for self-speculative decoding as MCTS over a zone- and group-structured action space is a fresh perspective that marries on-the-fly search with direct speedup feedback.
  - The zone-specific action design (execute-only in early zone; execute/skip in middle; execute/repeat in final) is a neat inductive bias that reduces search burden and encodes domain priors.
2. Experimental rigor and validation
  - Results on two model scales (8B and 70B) and five datasets with both greedy and temperature sampling provide a broad, task-diverse evaluation.
  - Ablations (w/o zone, w/o grouping, w/o MCTS) isolate the contributions of each component, and the latency breakdown indicates that the added search/initialization overhead is negligible compared to drafting and verification.
  - Additional analyses cover dynamic data streams, generation length sensitivity, and MCTS-iteration sensitivity.
3. Significance of contributions
  - A training-free, plug-and-play method that delivers near-2× speedups is practically meaningful for the community, especially given the difficulty of deploying trained draft models across varied settings.

Weakness
1. Experiment issues
  - Only batch size 1 is evaluated. Many serving scenarios benefit from batching; it remains unclear how LEAP behaves under batch>1 and long contexts (compute- vs memory-bound regimes).
  - Lack of comparison with other training-free speculative sampling methods, such as suffix decoding.
2. Presentation issues
  - Grouping strategy is underspecified (“GroupingStrategy”is a black box).

---

> ### Author Rebuttal · Authors · 2026-03-31
>
> **Note.** All experiments are conducted with **Temperature=0**.
>
> ----
>
> **[Q1]: How does LEAP scale under batch>1?**
>
> **[A1]**: We thank the reviewer for asking about LEAP’s scaling behavior under batch>1. We sincerely refer the reviewer to our **response Q4 to Reviewer xsr8** for a detailed discussion of this issue due to space constraint. Overall, LEAP remains beneficial beyond BS=1 and still outperforms CLaSP, though its speedup drops with larger batches due to higher batched overhead and no dedicated batch-serving optimization.
>
> ----
>
> **[Q2]: How does LEAP scale under longer contexts?**
>
> **[A2]**: We thank the reviewer for raising the evaluation of LEAP under longer-context setting. We sincerely refer the reviewer to our **response Q3 to Reviewer xsr8** for a detailed discussion of this issue. Overall, LEAP remains effective on long-context inputs, although its speedup is slightly reduced due to the increased overhead.
>
> ----
>
> **[Q3]: Lack of comparison with other training-free speculative sampling methods, such as suffix decoding.**
>
> **[A3]**: We thank the reviewer for this helpful suggestion. We agree that a broader comparison with other speculative decoding (SD) methods would better evaluate LEAP. For broader comparison, we evaluate LEAP against other **training-free SD methods**, including MTAD [1], DSBD [2], and Lookahead [3]. We refer the reviewer to our **response Q4 to Reviewer LySS** for a detailed discussion. In short, LEAP remains highly competitive among other training-free SD methods and is plug-and-play, making it easier to deploy in practice.
>
> We'd like to note that suffix decoding [4] is excluded because it targets a different workload: repetitive, agent-driven tasks, rather than the general non-agentic tasks. Therefore, a direct comparison would be unfair, since suffix decoding and other SD methods are optimized for different scenarios.
>
> -----
>
> **[Q4]: What is the exact definition and allowed multiplicity of the “repeat” action? How many repeats are permitted, and how frequently does LEAP select repeats in practice?**
>
> **[A4]**: We thank the reviewer for this question and apologize that the manuscript did not define the **repeat** action clearly enough. In LEAP, repeat is defined at the **layer-group level**. If a group contains layers \(A, B\),  applying repeat means executing this group **twice**, i.e., \(A, A, B, B\). In our implementation, repetition multiplicity is restricted to only one to better balance speed and performance, providing compensation for aggressive skipping while avoiding additional computation.
>
> In practice, repeat is **dynamically selected** by MCTS based on real-time feedback, and its usage frequency is **low**. As shown in **Figure 7 in Appendix C.4**, the usage frequency stays at or only slightly above 1, indicating that repetition is only an **optional** operation rather than a dominant choice to maintain the speed–performance trade-off, not as a frequently used action.
>
> ----
>
> **[Q5]: How is the grouping strategy determined (e.g., fixed group sizes, heuristic merging)?**
>
> **[A5]**:  We apologize that the grouping strategy was not clearly described in the manuscript. It is a deterministic, zone-aware heuristic based on the zone assignment together with $\Delta$EAR and $R_\ell$, and can be clarified from **group size** and **merging**.
>
> **For group size**, it is **not fixed globally**, but determined dynamically based on the size of each zone. Specifically, the maximum group size is set to $\max(3,\lfloor |Z_2|/6 \rfloor)$ for Zone 2 and $\max(2,\lfloor |Z_3|/4 \rfloor)$ for Zone 3, where $|Z_2|$ and $|Z_3|$ denote the numbers of layers assigned to Zone 2 and Zone 3, respectively. Therefore, when a zone contains more layers, slightly larger groups are allowed; when it is shorter, the groups remain compact.
>
> **For merging**, it is heuristic but fully explicit. We traverse layers from shallow to deep, and crossing a zone boundary always triggers a split. In **Zone 1**, all consecutive layers are merged into one group, since they only support the same action `EXECUTE`. In **Zone 2**, layers with unusually large $\Delta$EAR or sharply increased $R_\ell$ are treated as sensitive; transitions between sensitive and non-sensitive layers trigger a group split, while consecutive non-sensitive layers are merged until the group size is reached. In **Zone 3**, layers with top $\Delta$EAR values are also isolated, while remaining layers are merged unless a large increase in $\Delta$EAR appears or the adaptive group size is reached.
>
> ----
>
> **References:**
>
> [1] Qin, Zongyue, et al. Optimized multi-token joint decoding with auxiliary model for llm inference. 2024.
>
> [2] Qin, Zongyue, et al. Dynamic-width speculative beam decoding for llm inference. 2025.
>
> [3] Fu, Yichao, et al. Break the sequential dependency of llm inference using lookahead decoding. 2024.
>
> [4] Oliaro, Gabriele, et al. Suffixdecoding: Extreme speculative decoding for emerging ai applications. 2024.

---

> > ### Author Rebuttal · Reviewer_1Ecd · 2026-04-01
> >
> > The author answered my questions in detail and thoroughly explained the algorithm details and settings. I suggest that the author include these details in the article so that readers can understand how the system works. I will raise the score.

---

> > > ### Author Response · Authors · 2026-04-01
> > >
> > > We sincerely thank the reviewer for the positive and encouraging feedback! We truly appreciate your recognition that our response has clarified the algorithm details and settings more thoroughly. We fully agree that these details should be incorporated into the paper, and we will revise the manuscript accordingly so that readers can better understand how LEAP works. We are also very grateful for your willingness to raise the score.

---

### Official Review · Reviewer_xsr8 · 2026-03-10

**Soundness:** 3
**Presentation:** 3
**Significance:** 3
**Originality:** 3
**Overall Recommendation:** 4
**Confidence:** 4

**Summary:**

The paper proposes LEAP, a plug‑and‑play self‑speculative decoding method that constructs a draft model from a subset of target layers. It formulates draft‑layer selection as a sequential decision process solved by MCTS, guided by two empirical observations: prefilling‑derived redundancy remains informative during decoding, and redundancy exhibits zone‑wise structure. LEAP partitions layers into zones and groups to shrink the search space, then uses MCTS to find high‑speedup configurations. Experiments on LLaMA‑3‑8B/70B show ~1.7–2.0× wall‑time speedup with lossless verification.

**Compliance With Llm Reviewing Policy:**

Affirmed.

**Final Justification:**

Thank you for the detailed rebuttal. My concerns are fully resolved. I will rise my score.

**Key Questions For Authors:**

- Can you provide explicit statistics on the effective draft size (retained layers, draft FLOPs, or draft/verify latency split) for the MCTS‑selected configurations?
- How does LEAP compare to EAGLE‑3 (or similar SOTA lightweight drafts) under matched hardware and acceptance rate? If excluded, why is self‑spec preferable in practice?
- Do the gains persist in multi‑batch or longer‑context settings where verification costs and draft overheads differ?

**Limitations:**

yes

**Strengths And Weaknesses:**

Strengths
- Addresses a practical bottleneck in self‑speculative decoding: how to select draft layers efficiently and adaptively.
- The zone‑aware grouping provides a reasonable inductive bias that makes MCTS feasible under limited search budgets.
- Lossless verification ensures output distribution matches the target model, simplifying evaluation.

Weaknesses
- The paper does not report the effective draft size or compute share (e.g., average retained layers, draft FLOPs, or latency split between draft vs. verification). This makes it hard to assess the real cost of self‑spec.
- Comparisons are limited to self‑spec baselines (SWIFT, CLaSP). There is no comparison with SOTA lightweight drafts such as EAGLE‑3, which are common in real deployments.
- Reported speedups are moderate (≈2×) and are evaluated with batch size 1 and max length 1024; broader deployment regimes (multi‑batch, longer contexts) are not covered.

---

> ### Author Rebuttal · Authors · 2026-03-31
>
> **Note.** All experiments are conducted with **Temperature=0**.
>
> ---
>
> **[Q1]: Can you provide explicit statistics on the effective draft size for the MCTS‑selected configurations?**
>
> **[A1]**: We thank the reviewer for this suggestion. For **latency split**, we note that **Table 3 of manuscript** already reported the wall-clock latency breakdown of LEAP. Importantly, the newly introduced components together account for only negligible **0.1%** of total latency and the runtime is still dominated by draft and verification.
>
> For **retained layers**, LEAP determines it through online MCTS instead of fixing it as a hyper-parameter. On LLaMA-3-8B over GSM8K, the draft model executes **23/32 layers**  **(72% retention ratio)**. LEAP doesn't minimize the ratio to decrease the draft cost but instead targets the trade-off between draft cost and quality.
>
> ----
>
> **[Q2]: How does LEAP compare to EAGLE‑3 (or similar SOTA lightweight drafts)? If excluded, why is self‑spec preferable in practice?**
>
> **[A2]**: We sincerely thank the reviewer for this important question.
>
> We agree that EAGLE-3 is a strong baseline, but exclude it because it is **training-based**, whereas LEAP is **training-free and plug-and-play**. Since EAGLE-3 is also target-dependent, it may require extra training for unsupported or newly updated target models, and such overhead may be **mismatched** with users’ on-demand acceleration need.
>
> We believe self-speculative decoding remains preferable in practice because:
>
> - It requires **no additional training and no target-dependent drafter**. LEAP can build a drafter **online during inference**, making deployment simple and efficient.
>
> - Because drafting and verification happen within the same backbone, LEAP can **share the KV cache**, reducing memory footprint.
>
> For broader comparison, we also evaluate LEAP against other **training-free SD methods**, including MTAD [1], DSBD [2], and Lookahead [3]. We sincerely refer the reviewer to our **response Q4 to Reviewer LySS** for a detailed discussion due to space constraint. In short, LEAP remains highly competitive among training-free SD methods and is plug-and-play, making it easier and more efficient to deploy in practice.
>
> ---
>
> **[Q3]: Do the gains persist in longer‑context settings?**
>
> **[A3]**: We thank the reviewer for raising the evaluation of LEAP under longer-context setting. Regarding long-context task, we conduct experiments with Llama-3-8B on **LongBench_v2 with input lengths {8k, 32k, 64k, 128k}** .
>
> |   Dataset    |      8k      |     32k      |     64k      |     128k     |
> | :----------: | :----------: | :----------: | :----------: | :----------: |
> | LongBench_v2 | 1.87$\times$ | 1.77$\times$ | 1.85$\times$ | 1.85$\times$ |
>
> Results demonstrate that LEAP's speedup is slightly degraded on long-context inputs, suggesting its limitation in the long-text regime. As context length increases, the overhead becomes heavier in end-to-end latency to degrade the overall speedup. Nevertheless, LEAP still provides considerable speedups across long-context settings, indicating that it remains beneficial in the long-text regime.
>
> ----
>
> **[Q4]: Do the gains persist in multi‑batch settings where verification costs and draft overheads differ?**
>
> **[A4]:** We thank the reviewer for raising this important question. We agree that evaluating LEAP only at batch-size-1 is limited for understanding its behavior in practical deployment. We therefore evaluate LEAP on GSM8K using Llama-3-8B under larger batch sizes.
>
> | Method | BS=1  | BS=2  | BS=4  | BS=8  |
> | :----: | :---: | :---: | :---: | :---: |
> |  LEAP  | 2.02× | 1.87× | 1.60× | 1.41× |
> | CLaSP  | 1.47× | 1.49× | 1.36× | 1.13× |
>
> Results show that LEAP remains **beneficial and outperforms** CLaSP beyond batch-size-1, although its speedup gradually decreases with batch size. A similar decrease trend is also observed for CLaSP. The trend is expected: with larger batch sizes, both drafting and verification become heavier, while samples in the same batch may differ in token acceptance behavior, making batched speculation less favorable.
>
> We'd like to note that LEAP is designed as an algorithmic self-speculative method rather than a batch-serving optimization framework. Like CLaSP, it doesn't include specialized algorithms designed for large-batch serving, such as batch-level draft token selection in TETRIS [1] or KV-compressed drafting in MagicDec [2], which also helps explain why the speedup decreases as batch size grows.
>
> ---
>
> **References:**
>
> [1] Wu, Zhaoxuan, et al. TETRIS: Optimal draft token selection for batch speculative decoding. 2025.
>
> [2] Sadhukhan, Ranajoy, et al. Magicdec: Breaking the latency-throughput tradeoff for long context generation with speculative decoding. 2024.

---

> > ### Author Rebuttal · Reviewer_xsr8 · 2026-04-03
> >
> > The reported 72% layer retention ratio indicates that the draft model is far from lightweight. By contrast, methods   like EAGLE-3 achieve a draft-to-target parameter ratio of roughly 1:5 on 8B models and as favorable as 1:10–20 on larger architectures like 235B. With 72% of layers retained, the draft stage itself already consumes the majority of target-model compute, leaving minimal room for net speedup once other speculative overhead is accounted for, which raises concerns about whether this approach can yield meaningful acceleration on larger models at all.
> >
> > Regarding Q2, the supplementary comparison provided in the response to Reviewer LySS is limited to other training-free    methods (MTAD, DSBD, Lookahead), which does not address my original concern — in practical deployment, users choose   among all available acceleration approaches, and training-based methods like EAGLE-3 represent realistic alternatives   that should be compared against to contextualize the actual value of the self-speculative paradigm.

---

> > > ### Author Response · Authors · 2026-04-03
> > >
> > > **[Q1]: The reported 72% layer retention ratio indicates that the draft model is far from lightweight. By contrast, methods like EAGLE-3 achieve a draft-to-target parameter ratio of roughly 1:5 on 8B models and as favorable as 1:10–20 on larger architectures like 235B. With 72% of layers retained, the draft stage itself already consumes the majority of target-model compute, which raises concerns about whether this approach can yield meaningful acceleration on larger models at all.**
> > >
> > > **[A1]**: We thank the reviewer for this thoughtful comment. We first clarify that the retained-layer ratio alone is not sufficient to judge the practical value of a self-speculative method. As reflected in Speculative Decoding [1], the **speedup gain** is
> > > $$
> > > \frac{1-\alpha^{\gamma+1}}{(1-\alpha)(\gamma c+1)},
> > > $$
> > > so the final acceleration depends jointly on **draft length** $\gamma$, **draft cost** $c$, and **acceptance rate** $\alpha$, rather than on draft cost alone. In other words, making the drafter smaller is not always better: if too many layers are skipped, the drafting cost may decrease, but the acceptance rate can also drop, which may even degrade the final speedup. Therefore, the retained-layer ratio should be viewed as the optimal result of the **speedup trade-off**, rather than an objective to be minimized.
> > >
> > > We also agree that EAGLE-3 uses a much smaller drafter. However, this lightweight drafter is not obtained for free: its ability to remain aligned with the target model relies on **additional, target-specific training**. By contrast, LEAP does not require any target-specific training or even the availability of a smaller compatible model from the same family. This makes self-speculative decoding substantially easier to apply in practice, especially when the target model is unsupported, updated, private, or when users need **on-demand acceleration**.
> > >
> > > Finally, we would like to emphasize that LEAP already shows meaningful speedups across multiple model scales. In particular, we observe consistent acceleration on **Llama-3.2-1B, Llama-3-8B and Llama-3-70B**, which demonstrates that LEAP can deliver practical speedup across different model sizes. Therefore, our results support that LEAP remains a meaningful and practically useful solution in the **training-free self-speculative** setting.
> > >
> > > ----
> > >
> > > **[Q2]: In practical deployment, users choose among all available acceleration approaches, and training-based methods like EAGLE-3 represent realistic alternatives that should be compared against to contextualize the actual value of the self-speculative paradigm.**
> > >
> > > **[A2]**: We thank the reviewer for this valuable suggestion. We agree that, in practical deployment, users often choose among all available SD methods, and training-based approaches such as **EAGLE-3** are indeed available choices. We therefore add a direct comparison with **EAGLE-3** on **Llama-3.1-8B-Instruct**.
> > >
> > > | Method  | Immediate Use Without Extra Training | GSM8K | CNN/DM | WMT14_DEEN |  NQ   | MT-Bench | Overall |
> > > | :-----: | :----------------------------------: | :---: | :----: | :--------: | :---: | :------: | :-----: |
> > > |  LEAP   |                 Yes                  | 1.89× | 1.71×  |   1.87×    | 1.89× |  1.84×   |  1.84×  |
> > > | EAGLE-3 |                  No                  | 2.99× | 2.48×  |   2.50×    | 2.59× |  1.75×   |  2.44×  |
> > >
> > > From the table, we observe that **EAGLE-3 achieves higher absolute speedups** than LEAP. We believe this is expected, since EAGLE-3 relies on an additional **trained drafter** which provides extra optimization capacity beyond self-speculative decoding. However, this gap reflects a fundamental difference in practical trade-offs rather than strict superiority.
> > >
> > > Specifically, although EAGLE-3 is indeed an available option in practical deployment, its stronger speedup is achieved at the cost of **additional training**. In practice, such a requirement is **not always available or affordable**. Although EAGLE releases checkpoints for multiple target models, **coverage is still limited**, and many target models are not supported by public checkpoints. In such cases, users must train an EAGLE drafter themselves for the target model using large amounts of training data, which can substantially **limit its applicability** to some extent.
> > >
> > > By contrast, LEAP requires neither an auxiliary drafter nor target-alignment training, and can be applied immediately once the target model is available. This makes LEAP particularly beneficial in scenarios where users need a **training-free, plug-and-play** SD solution with limited computational resource.
> > >
> > > Overall, this comparison shows that EAGLE-3 and LEAP reflect **different practical trade-offs**: EAGLE-3 pursues higher speedup with extra training, while LEAP emphasizes training-free usability, simplicity, and lower deployment cost.
> > >
> > > ----
> > >
> > > [1] Leviathan, Yaniv, Matan Kalman, and Yossi Matias. Fast inference from transformers via speculative decoding. 2023.

---

### Official Review · Reviewer_LySS · 2026-03-12

**Soundness:** 2
**Presentation:** 3
**Significance:** 3
**Originality:** 3
**Overall Recommendation:** 4
**Confidence:** 3

**Summary:**

This paper studies lossless self-speculative decoding for LLM inference by constructing the draft model from subsets of the target model’s own layers and optimizing that construction online with a zone-aware MCTS procedure. The core idea is to use prefilling-derived redundancy signals to partition layers into early, middle, and final zones, then search over group-level execute/skip/repeat decisions during decoding. The paper shows consistent inference gains over prior plug-and-play self-speculative baselines across LLaMA-3-8B and 70B, with reported overall speedups around 1.7–2.0x while preserving the target model’s output distribution through standard lossless verification.

**Compliance With Llm Reviewing Policy:**

Affirmed.

**Final Justification:**

The rebuttal has partially resolved my concerns, thus improving the score.

**Key Questions For Authors:**

How does LEAP behave under different deployment regimes, especially larger batch sizes, continuous batching, or other hardware settings?

**Limitations:**

The paper does not sufficiently discuss limitations. I would suggest the authors to briefly discuss the potential sensitivity to hyperparameters, limited evaluation across models and deployment settings, and possible variability of performance across different prompts or inference workloads.

**Strengths And Weaknesses:**

Pros:
1. The paper addresses an important systems challenge in LLM inference, reducing decoding latency without adding auxiliary models.
2. The proposed zone-aware MCTS framework is an interesting approach to optimizing self-speculative decoding. By structuring the search space using layer redundancy patterns and grouping layers into zones, the authors significantly reduce the otherwise intractable configuration space.
3. The paper reports consistent speedups across multiple datasets and model sizes (LLaMA-3-8B and 70B), outperforming prior plug-and-play self-speculative baselines such as SWIFT and CLaSP. The reported 1.7–2.0× acceleration is meaningful in practice.

Cons:
1. The method relies heavily on empirical observations (e.g., redundancy stability and zone-wise behavior) without a strong theoretical explanation for why these properties should consistently hold across prompts, tasks, or model architectures.
2. The paper compares mainly against two plug-and-play self-speculative baselines. A broader discussion or empirical comparison with other speculative decoding approaches would help better position the work within the overall literature.
3. The evaluation focuses on batch-size-1 inference. It is unclear how the approach behaves under realistic serving conditions such as larger batch sizes.

---

> ### Author Rebuttal · Authors · 2026-03-31
>
> **Note.** All experiments below are conducted with **Temperature=0**.
>
> ----
>
> **[Q1]: The evaluation focuses on batch-size-1 inference.**
>
> **[A1]**: We thank the reviewer for this important question. We sincerely refer the reviewer to our **response Q4 to Reviewer xsr8** for a detailed discussion due to space constraint. Overall, LEAP remains beneficial beyond BS=1 and still outperforms CLaSP, though its speedup drops with larger batches due to higher batched overhead and no dedicated batch-serving optimization.
>
> ---
>
> **[Q2]: The paper does not sufficiently discuss limitations.**
>
> **[A2]**: We thank the reviewer for highlighting this issue. We clarify LEAP’s limitations in hyperparameter sensitivity, smaller models, and long inputs.
>
> - For hyperparameter sensitivity, we evaluate **$\lambda$ and $c$ in Eq. (5)** on Llama-3-8B. The results show only moderate degradation under non-default settings, suggesting that LEAP is reasonably robust to hyperparameters.
>
>   |   $\lambda$   |    GSM8K     |   MT-Bench   |
>   | :-----------: | :----------: | :----------: |
>   |       0       | 1.85$\times$ | 1.88$\times$ |
>   | 0.3 (default) | 2.02$\times$ | 2.03$\times$ |
>   |      0.5      | 1.87$\times$ | 1.91$\times$ |
>   |      1.0      | 1.88$\times$ | 1.87$\times$ |
>
>   |     $c$     |    GSM8K     |   MT-Bench   |
>   | :---------: | :----------: | :----------: |
>   |      1      | 1.99$\times$ | 1.89$\times$ |
>   |      2      | 1.82$\times$ | 1.85$\times$ |
>   | 4 (default) | 2.02$\times$ | 2.03$\times$ |
>   |      8      | 1.80$\times$ | 1.87$\times$ |
>
> - For smaller models, we conduct experiments on **Llama-3.2-1B (16 layers)**. LEAP remains effective and competitive, though speedup degrades due to weaker layer heterogeneity and model redundancy.
>
>   | Method |      GSM8K       |     MT-Bench     |      CNNDM       |
>   | :----: | :--------------: | :--------------: | :--------------: |
>   |  LEAP  | **1.58$\times$** | **1.70$\times$** | **1.52$\times$** |
>   | CLaSP  |   1.33$\times$   |   1.26$\times$   |   1.18$\times$   |
>
> - For long-context tasks, we conduct experiments on LongBench_v2 with **input lengths {8k, 32k, 64k, 128k}**. We sincerely refer the reviewer to our **response Q3 to Reviewer xsr8** for a detailed discussion.
>
> ---
>
> **[Q3]: The method relies heavily on empirical observations (e.g., redundancy stability and zone-wise behavior).**
>
> **[A3]**: We thank the reviewer for this insightful comment. We'd like to note that heuristic priors are common in **self-speculative** methods. For example, CLaSP uses token feature alignment heuristics, while SWIFT treats layer selection as a black-box problem.
>
> We also agree that LEAP is empirically motivated, and that redundancy stability and zone-wise behavior may not hold equally well in all settings. However, in our studied settings, they appear stable enough to provide useful structural priors. Our preliminary analysis shows clear zone-wise patterns, and the main results suggest that these priors help construct an effective search space and yield consistent acceleration.
>
> Importantly, these metrics and zone partitioning do not directly determine the final speedup. They only initialize the search space; the final layer configuration is still optimized online by MCTS with real-time feedback. This is also reflected in the **Llama-3.2-1B** results, where LEAP still achieves considerable speedup despite coarser partitioning.
>
> ----
>
> **[Q4]: Lack of a broader comparison with other SD methods.**
>
> **[A4]**: We thank the reviewer for this helpful suggestion. To broaden the comparison, we evaluate LEAP against other **training-free SD methods**, including MTAD [3], DSBD [4], and Lookahead [5].
>
> | Dataset  |       LEAP       |       MTAD       |     DSBD     |  Lookahead   |
> | :------: | :--------------: | :--------------: | :----------: | :----------: |
> |  GSM8K   | **2.02$\times$** |   1.95$\times$   | 1.58$\times$ | 1.87$\times$ |
> | MT-Bench |   2.03$\times$   | **2.34$\times$** | 1.59$\times$ | 1.85$\times$ |
>
> On GSM8K, LEAP achieves the best speedup. On MT-Bench, LEAP outperforms DSBD and Lookahead but not MTAD, likely because MTAD uses **lossy verification**, which improves acceptance but does not preserve the target output distribution. LEAP instead focuses on **lossless acceleration**, reflecting its trade-off between speedup and quality.
>
> Overall, LEAP remains highly competitive among training-free non-self-speculative methods. More importantly, it is **lossless and training-free**, easing practical deployment.
>
> ---
>
> **References:**
>
> [1] Qin, Zongyue, et al. Optimized multi-token joint decoding with auxiliary model for llm inference. 2024.
>
> [2] Qin, Zongyue, et al. Dynamic-width speculative beam decoding for llm inference. 2025.
>
> [3] Fu, Yichao, et al. Break the sequential dependency of llm inference using lookahead decoding. 2024.

---

> > ### Author Rebuttal · Reviewer_LySS · 2026-04-03
> >
> > Thank you for the thoughtful rebuttal, especially the added batch-size experiments, which are helpful. It’s great to see that the gains persist beyond BS=1. I still have some mild questions around deployment realism (e.g., continuous batching or batch-aware optimizations), and how LEAP might integrate into such settings.
> >
> > Regarding comparisons, the additional baselines are helpful. It may further strengthen the paper to more clearly position LEAP relative to a broader range of approaches, including training-based methods (e.g., Eagle3), to better highlight its practical trade-offs.

---

> > > ### Author Response · Authors · 2026-04-04
> > >
> > > **[Q1]: I still have some mild questions around deployment realism (e.g., continuous batching or batch-aware optimizations), and how LEAP might integrate into such settings.**
> > >
> > > **[A1]**: We thank the reviewer for this helpful follow-up. We agree that more realistic deployment settings, such as continuous batching, are important for strengthening the practical applicability of LEAP. At the same time, we agree that this is better viewed as **an important extension of LEAP**, rather than a core limitation of its current algorithmic design.
> > >
> > > **Conceptually**, LEAP is orthogonal to large-batch algorithm such as continuous batching or batch-aware optimization. A concrete integration would be to use **LEAP** as the per-request self-speculative method to construct the draft model, while applying **continuous batching** to dynamically manage request insertion or removal. However, this is **not entirely plug-and-play**, since continuous batching introduces requests with different lengths and decoding progress, and LEAP further adds request-specific draft/verification. As a result, **additional adaptations** are still needed, such as scheduling policies to coordinate requests with different decoding progress, and KV-cache management to handle dynamic batch composition and asynchronous progress.
> > >
> > > We agree that a full empirical study under continuous batching would further strengthen this point, but we view this integration as a larger system-level extension **beyond the scope of the short rebuttal**, and we also view it as a promising direction for our future work.
> > >
> > > ----
> > >
> > > **[Q2]: It may further strengthen the paper to more clearly position LEAP relative to a broader range of approaches, including training-based methods (e.g., Eagle3), to better highlight its practical trade-offs.**
> > >
> > > **[A2]**: We thank the reviewer for this valuable follow-up. We agree that a comparison with training-based methods such as EAGLE-3 is useful for broader positioning and for clarifying the practical trade-offs of LEAP. In response, we have added this comparison on **Llama-3.1-8B-Instruct**.
> > >
> > > | Method  | Training-free | GSM8K | CNN/DM | WMT14_DEEN |  NQ   | MT-Bench | Overall |
> > > | :-----: | :-----------: | :---: | :----: | :--------: | :---: | :------: | :-----: |
> > > |  LEAP   |      Yes      | 1.89× | 1.71×  |   1.87×    | 1.89× |  1.84×   |  1.84×  |
> > > | EAGLE-3 |      No       | 2.99× | 2.48×  |   2.50×    | 2.59× |  1.75×   |  2.44×  |
> > >
> > > We observe that **EAGLE-3 achieves higher absolute speedup** than LEAP. We believe this is expected, since EAGLE-3 relies on an additional **trained drafter**, which provides extra optimization capacity beyond self-speculative decoding. This performance gap, however, does not diminish the value of LEAP, because the two methods target **different practical trade-offs**.
> > >
> > > More specifically, EAGLE-3 achieves stronger speedup at the cost of additional training, along with additional training time and deployment overhead. LEAP, in contrast, is designed as a **training-free, plug-and-play self-speculative method** that can be applied immediately once the target model is available. In this sense, LEAP prioritizes a different point in the trade-off: lower deployment overhead and broader usability, rather than the strongest absolute speedup.
> > >
> > > We also note that our original experiment focuses on comparing **different ways of constructing the draft model within the self-speculative paradigm**, rather than against methods that rely on trained external drafters. That is why EAGLE-3 is not included in the manuscript initially. Nevertheless, we agree that it is an **important stronger baseline** for broader comparison, and this additional result helps position LEAP more clearly within the overall SD literature.
> > >
> > > Overall, we believe this comparison better highlights the practical trade-off: **EAGLE-3 offers higher speedup, while LEAP offers a training-free and simpler deployment**, highlighting their different trade-offs between speedup and deployment simplicity.

---

### Decision · Program_Chairs · 2026-04-30

**Decision:**

Accept (regular)

**Comment:**

The work addresses a timely question in self-speculative decoding, and the reviewers identified the key strengths of this work as
1) The zone‑aware grouping is a novel and well-motivated inductive bias.
2) The evaluation is conducted over diverse model scales and data sets.
3) Speed-ups with LEAP are demonstrated across the diverse experiments.  Additional experiments in the rebutal demonstrate that it compares favorably with a variety of existing baselines.

The reviewers noted a few weaknesses/limitations of the existing presentation.
1)  Many reviewers noted that the experiments were only done with batch size 1.  While the authors did provide numbers in their rebuttal, these results should be incorporated into the main text, and they should be paired with computational costs to put them in perspective.
2)  The reviewers also suggested a variety of additional baselines, e.g., EAGLE-3.  Again, these comparisons were provided in the rebuttal.  Many of the reviewers agreed that these results would strengthen the arguments of paper.
3)  The work is highly empirical and lacks a rigorous theoretical justification.

Overall, the reviewers agreed that the novelty of the approach and empirical performance of the LEAP method are likely of interest to the speculative decoding community.